# Neural SDF Flow for 3D Reconstruction of Dynamic Scenes

**Wei Mao**
Australian National University
`wei.mao@anu.edu.au`

**Richard Hartley**
Australian National University & Google
`richard.hartley@anu.edu.au`

**Mathieu Salzmann**
CVLab, EPFL & SDSC, Switzerland
`mathieu.salzmann@epfl.ch`

**Miaomiao Liu**
Australian National University
`miaomiao.liu@anu.edu.au`

## Abstract

In this paper, we tackle the problem of 3D reconstruction of dynamic scenes from multi-view videos. Previous dynamic scene reconstruction works either attempt to model the motion of 3D points in space, which constrains them to handle a single articulated object or require depth maps as input. By contrast, we propose to directly estimate the change of Signed Distance Function (SDF), namely SDF flow, of the dynamic scene. We show that the SDF flow captures the evolution of the scene surface. We further derive the mathematical relation between the SDF flow and the scene flow, which allows us to calculate the scene flow from the SDF flow analytically by solving linear equations. Our experiments on real-world multi-view video datasets show that our reconstructions are better than those of the state-of-the-art methods. Our code is available at `https://github.com/wei-mao-2019/SDFFlow.git`.

## 1 Introduction

The 3D reconstruction of a dynamic scene from multi-view videos is a very challenging research topic compared to its counterpart for static scenes. Yet, it has many important applications ranging from virtual/augmented reality to videos games, where it is required to model changes in the 3D environment, i.e., surface deformations. To handle such deformations, traditional non-rigid structure from motion methods (Bregler et al., 2000; Akhter et al., 2008) require 2D correspondences across time. While more recent works (Pumarola et al., 2021; Park et al., 2021a; Li et al., 2021) tackle this problem with neural rendering techniques, i.e., NeRF (Mildenhall et al., 2021), almost all those works (Pumarola et al., 2021; Park et al., 2021a; Li et al., 2021) directly model the movements of 3D points. Despite their great success, these methods mainly focus on synthesizing photo-realistic novel views and cannot obtain good 3D geometry due to the shape radiance ambiguity (Zhang et al., 2020). To resolve such ambiguity, a commonly used strategy is to parameterize the density with Signed Distance Function (SDF) (Wang et al., 2021b; Yariv et al., 2021).

In this work, we aim for the reconstruction of an *unconstrained* dynamic scene and for recovering the 3D motion of the scene, i.e., scene flow, using NeRF (Mildenhall et al., 2021). Previous works (Yang et al., 2022; Xu et al., 2021; Wang et al., 2022; Grassal et al., 2022; Hong et al., 2022; Guo et al., 2023) on dynamic object reconstruction are restricted to either a single articulated object such as a human (Yang et al., 2022; Xu et al., 2021; Wang et al., 2022; Guo et al., 2023) or a pre-defined template such as human head (Grassal et al., 2022; Hong et al., 2022). To handle an *unconstrained* scene that may contain multiple (non-)rigid objects, existing NeRF-based methods (Cai et al., 2022; Shao et al., 2023) either require mapping the 3D points to a higher dimensional space to account for topology changes (Cai et al., 2022) or directly predict the SDF at each time step (Shao et al., 2023), thus preventing them to recover the scene flow.

By contrast, we propose a novel representation, namely SDF flow, which naturally captures the topology changes and allows us to infer the scene flow. Based on the observation that the SDF of any given point in a dynamic scene is continuous and almost everywhere smooth with respect to

time, our SDF flow is defined as the first-order derivative of the SDF with respect to time. Given such an SDF flow, the SDF at any point and at any given time is simply the integral of its SDF flow. We then develop a NeRF-based method that, instead of directly predicting the SDF, is trained to estimate the SDF flow, allowing us to extract the 3D scene motion.

To obtain such a 3D scene motion, we derive the mathematical relationship between the SDF flow and the scene flow. Specifically, we show that the SDF flow of a 3D point can be expressed as a linear function of its location, surface normal, and the scene flow. We then demonstrate that, without any supervision, we can analytically compute the scene flow from the SDF flow. Although such scene flow can be noisy due to the reconstruction error, we showcase that the linear relationship provides a good regularization on both the scene flow and SDF flow resulting better scene flow estimation and 3D reconstruction. We believe that revealing this relationship will be valuable for future research.

Our contributions can be summarized as follows: i) we propose the SDF flow as a novel 3D representation of *unconstrained* dynamic scenes; ii) we unify the SDF flow and the scene flow with a linear function; iii) With our SDF flow representation, we introduce a NeRF-based pipeline that can reconstruct the 3D geometry of a dynamic scene given multi-view videos. We evaluate our method on real world multi-view videos, and our model reconstructs more accurate surfaces than those of the state-of-the-art dynamic scene reconstruction methods.

## 2 RELATED WORK

**Neural radiance field for dynamic scenes.** Given multi-view images, a neural radiance field (NeRF) (Mildenhall et al., 2021) optimizes a continuous function that maps any 3D location to its density and radiance. Such a function has been proven to be effective for novel view synthesis of static scenes. Recent works (Park et al., 2021a;b; Pumarola et al., 2021; Li et al., 2021; Tretschk et al., 2021; Du et al., 2021; Wang et al., 2021a; Song et al., 2022; Fang et al., 2022; Li et al., 2022; 2023) further extend it to dynamic scenes. Most of them propose to optimize additional functions that deform the observed points to a canonical space (Park et al., 2021a;b; Pumarola et al., 2021; Tretschk et al., 2021; Fang et al., 2022) or over time (Li et al., 2021; Wang et al., 2021a; Du et al., 2021; Li et al., 2023). Despite their good quality of novel view synthesis, these methods cannot reconstruct faithful 3D scene geometry due to the "shape-radiance ambiguity" (Zhang et al., 2020).

**3D reconstruction of dynamic scenes.** Traditional non-rigid structure from motion (NRSfM) methods (Bregler et al., 2000; Akhter et al., 2008) reconstruct deformable 3D shapes from a set of 2D correspondences. Such correspondences are sometimes hard to obtain, making these methods not suitable to complex real-world scenes. Although some works (Blanz et al., 2003; Cao et al., 2014; Ichim et al., 2015; Thies et al., 2016; Guo et al., 2018; Gafni et al., 2021; Yang et al., 2021a;b; Xu et al., 2021; Yang et al., 2022; Wang et al., 2022; Hong et al., 2022; Grassal et al., 2022; Guo et al., 2023) can reconstruct non-rigid objects without requiring 2D correspondences, they assume the reconstructed object to be either articulated or follow certain pre-defined templates such as, Cao et al. (2013); Li et al. (2017); Blanz & Vetter (2023). Such assumptions make these methods not suitable for unconstrained scenes where there may be multiple non-rigid moving objects. Other works (Newcombe et al., 2015; Innmann et al., 2016; Slavcheva et al., 2017; Lin et al., 2022) that can handle unconstrained scenes, require depth map as inputs. Two existing NeRF-based methods can nonetheless handle unconstrained scenes without depth information: NDR (Cai et al., 2022) and Tensor4D (Shao et al., 2023). NDR (Cai et al., 2022) introduces a bijective function that maps the points in observation space to a canonical space. It requires to extend the 3D input to a higher dimensional space to account for the topology changes (Park et al., 2021b). Tensor4D (Shao et al., 2023) represents the dynamic scene with a 4D tensor and further decomposes the tensor into several 2D planes to speed up training and inference. However, since their method directly estimate the SDF at each time step, it cannot recover the scene flow. Our SDF flow naturally captures the smooth deformations of the surface and handles topology changes by design. Given the SDF flow, we can further obtain the scene flow.

## 3 OUR APPROACH

In this section, we first briefly introduce the neural radiance field and the SDF-based parameterization of the density (Section 3.1). We then describe our SDF flow to capture the dynamic scenes (Section 3.2). Lastly, we derive the mathematical relationship between the SDF flow and the scene flow (Section 3.3).

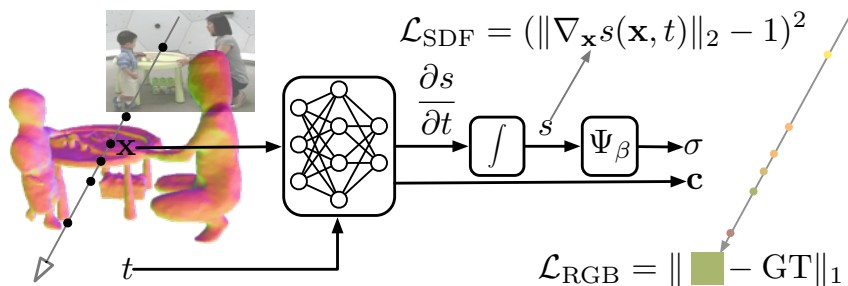

$$\mathcal{L}_{\text{SDF}} = (\|\nabla_{\mathbf{x}} s(\mathbf{x}, t)\|_2 - 1)^2$$

$$\mathcal{L}_{\text{RGB}} = \| \quad - \text{GT}\|_1$$

Figure 1: **Pipeline.** We propose the SDF flow $\frac{\partial s}{\partial t}$ to capture the deformation of a dynamic scene.

### 3.1 PRELIMINARIES

**Neural radiance field (NeRF).** The main idea of NeRF (Mildenhall et al., 2021) is to represent a static scene as a 5D continuous function that maps a 3D location $x \in \mathbb{R}^3$ and a viewing direction $d \in \mathbb{R}^2$ to the RGB radiance $c \in \mathbb{R}^3$ and the density $\sigma \in \mathbb{R}$, i.e.,

$$c, \sigma = f_\Theta(x, d) , \tag{1}$$

where the function $f$ is typically implemented as a Multi-Layer Perceptron (MLP) with $\Theta$ as trainable parameters.

Given such function, for each ray $r(r) = o + rd$ shooting from the camera origin $o \in \mathbb{R}^3$ along direction $d$, one can obtain the pixel intensity via the volume rendering function

$$\mathbf{C}(r) = \int_{r_n}^{r_f} T(r)\sigma(r(r))c(r(r), d)dr, \tag{2}$$

where $r_n$ and $r_f$ are the bound of the 3D scene; $T(r) = e^{-\int_{r_n}^{r} \sigma(r(l))dl}$ is the accumulated opacity; $\mathbf{C}(r) \in \mathbb{R}^3$ is the rendered color of this ray.

The model can then be trained by minimizing the loss between the rendered color $\mathbf{C}(r)$ and the ground-truth $\bar{\mathbf{C}}(r)$, i.e.,

$$\mathcal{L}_{\text{RGB}} = \|\mathbf{C}(r) - \bar{\mathbf{C}}(r)\|_1. \tag{3}$$

**Volume rendering with SDF.** It has been shown that NeRF may not recover the correct 3D geometry due to the shape radiance ambiguity (Zhang et al., 2020). To address this issue, a few works have proposed to regularize the density by parameterizing it as an SDF (Wang et al., 2021b; Yariv et al., 2021). Taking VolSDF (Yariv et al., 2021) as an example, the density $\sigma$ is defined as

$$\sigma = \frac{1}{\beta}\Psi_\beta(s(x)), \tag{4}$$

where $\Psi_\beta$ is the Cumulative Distribution Function (CDF) of the Laplace distribution with zero mean and scale $\beta$. Instead of directly estimating the density in Equation 1, the function $f$ outputs the SDF $s(x)$.

To extend NeRF to dynamic scenes, the most commonly adopted strategy is to jointly optimize an additional function that models the deformation in 3D space such function either maps all observation spaces to a canonical one such as Cai et al. (2022) or models the temporal motion of the scene such as Li et al. (2021). In the next section, we propose a drastically different representation, i.e., SDF flow, which tries to directly model the change of the dynamic scenes over time.

### 3.2 SDF FLOW

Let $\Omega \subset \mathbb{R}^3$ represent the 3D space occupied by a scene and $\partial\Omega$ be the scene surface. The Signed Distance Function (SDF) $s(x)$ is defined as $s(x) = \begin{cases} -\min_{y \in \partial\Omega} \|x - y\|_2 & \text{if } x \in \Omega \\ \min_{y \in \partial\Omega} \|x - y\|_2 & \text{otherwise.} \end{cases}$ .

For any given 3D point $x$ in a dynamic scene, we can treat its SDF $s(x, t)$ as a function of time $t$. As shown in Figure 2, such a function is always continuous and almost everywhere differentiable in the real world scenario. To model the continuous function, we propose to estimate its first-order

derivative $\frac{\partial s}{\partial t}$ (SDF flow) as $\frac{\partial s(\boldsymbol{x},t)}{\partial t} = f(\boldsymbol{x},t)$ , where $f$ is the function to be optimized during training.

The SDF of point $\boldsymbol{x}$ at time $t$ is then the integral of its SDF flow, i.e.,

$$s(\boldsymbol{x},t) = \int_{t_0}^{t} f(\boldsymbol{x},t)dt + s(\boldsymbol{x},t_0) \ , \tag{5}$$

where $s(\boldsymbol{x},t_0)$ is the SDF of $\boldsymbol{x}$ at the initial time $t_0$, which can be produced by another function as $s(\boldsymbol{x},t_0) = f_0(\boldsymbol{x})$. We can also obtain its normal as

$$\boldsymbol{n}(\boldsymbol{x},t) = \nabla_{\boldsymbol{x}} s = \int_{t_0}^{t} \nabla_{\boldsymbol{x}} f(\boldsymbol{x},t)dt + \boldsymbol{n}(\boldsymbol{x},t_0), \tag{6}$$

where $\boldsymbol{n}(\boldsymbol{x},t_0) = \nabla_{\boldsymbol{x}} s(\boldsymbol{x},t_0) = \nabla_{\boldsymbol{x}} f_0(\boldsymbol{x}) \in \mathbb{R}^3$.

Figure 1 provides an overview of our pipeline. Given the SDF $s(\boldsymbol{x},t)$ of point $\boldsymbol{x}$ at time $t$, we can compute its density $\sigma(s(\boldsymbol{x},t))$ using Equation 4. We follow the neural rendering pipeline to further optimize another function that produces the radiance $\boldsymbol{c}$. Given the density and the radiance, we use the volume rendering equation defined in Equation 2 to obtain the final rendered RGB color $\mathbf{C}(\boldsymbol{r},t)$.

Our training loss consists of two parts:

$$\mathcal{L} = \mathcal{L}_{\text{RGB}} + \lambda \mathcal{L}_{\text{SDF}}, \tag{7}$$

where $\mathcal{L}_{\text{RGB}} = \mathbb{E}_{\boldsymbol{r} \in \mathbb{P}, t \in \mathbb{T}} \|\mathbf{C}(\boldsymbol{r},t) - \bar{\mathbf{C}}(\boldsymbol{r},t)\|_1$ is the average color loss over all sampled rays $\mathbb{P}$ across all times $\mathbb{T}$, and $\mathcal{L}_{\text{SDF}} = \mathbb{E}_{\boldsymbol{x} \in \mathbb{X}, t \in \mathbb{T}}(\|\boldsymbol{n}(\boldsymbol{x},t)\|_2 - 1)^2$ is the eikonal constraint of the SDF for all sampled points $\mathbb{X}$ across all times $\mathbb{T}$. $\lambda$ is a balancing weight.

Since it is often beneficial to obtain the 3D correspondences described by the scene flow for many applications or downstream tasks, in the next section, we derive the mathematical relation between the proposed SDF flow and the scene flow.

### 3.3 RELATION BETWEEN SDF FLOW AND SCENE FLOW

Given any point $\boldsymbol{x}$ on the surface $\partial\Omega$, we first define its $\epsilon$-neighbor as a local region on the surface that contains that point, i.e., $\mathcal{N}_\epsilon(\boldsymbol{x}) = \{\boldsymbol{y} | \|\boldsymbol{y} - \boldsymbol{x}\|_2 < \epsilon, \boldsymbol{y} \in \partial\Omega, \boldsymbol{x} \in \partial\Omega\}$ .

When considering the scenario where the surface is evolving within time $\Delta t > 0$, we make the following assumption and derive a theorem, which we will illustrate via a toy example at the end of the section.

**Assumption 1** *As time period $\Delta t$ approaches zero, with sufficiently small $\epsilon$, the motion of a surface point $\boldsymbol{x}$'s $\epsilon$-neighbor $\mathcal{N}_\epsilon(\boldsymbol{x})$ is rigid and can be represented as a rotation $\Delta\mathbf{R} \in SO(3)$ and a translation $\Delta\mathbf{T} \in \mathbb{R}^3$ . Thus, we can obtain the corresponding location $\boldsymbol{x}'$ of point $\boldsymbol{x}$ after $\Delta t$ as*

$$\boldsymbol{x}' = \Delta\mathbf{R}\boldsymbol{x} + \Delta\mathbf{T} \ . \tag{8}$$

**Theorem 2** *Given a 3D location $\boldsymbol{x}$ which is on a locally smooth surface deforming smoothly, the SDF change of $\boldsymbol{x}$ thus the first order derivative of its SDF with respect to time is the negative projection of its scene flow on to its normal. Specifically,*

$$\frac{\partial s}{\partial t} = \lim_{\Delta t \to 0} \frac{\Delta s}{\Delta t} \tag{9}$$

$$= -\frac{\partial \boldsymbol{x}}{\partial t}^T \boldsymbol{n}(\boldsymbol{x}) \ , \tag{10}$$

*where $\frac{\partial \boldsymbol{x}}{\partial t} \in \mathbb{R}^3$ is the scene flow and $\boldsymbol{n}(\boldsymbol{x}) \in \mathbb{R}^3$ is the surface normal at location $\boldsymbol{x}$.*

We provide the proof in Section A.1 and illustrate the proof in Figure 3.

Combining the Assumption 1 and Theorem 2, we have

$$\frac{\partial s}{\partial t} = -(\boldsymbol{\omega} \times \boldsymbol{x} + \boldsymbol{v})^T \boldsymbol{n}(\boldsymbol{x}) \ , \tag{11}$$

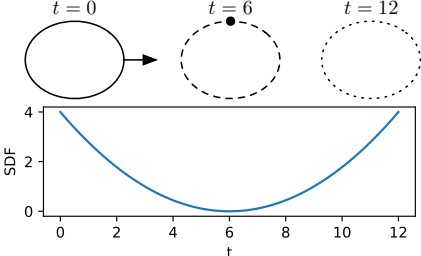

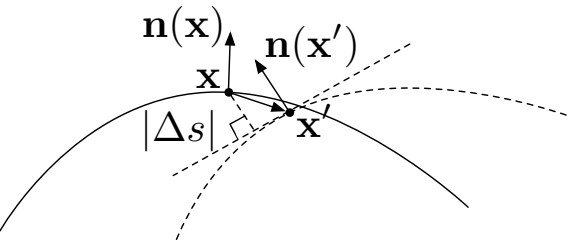

Figure 2: Considering an ellipse moving right at constant speed (top), the SDF of the point as a function of time is always differentiable (bottom).

Figure 3: 2D example of the relation between the scene flow $(\boldsymbol{x}' - \boldsymbol{x})$ and the SDF flow $\Delta s$. The solid curve is the surface around $\boldsymbol{x}$. The dashed one is the deformed surface after a very short time period.

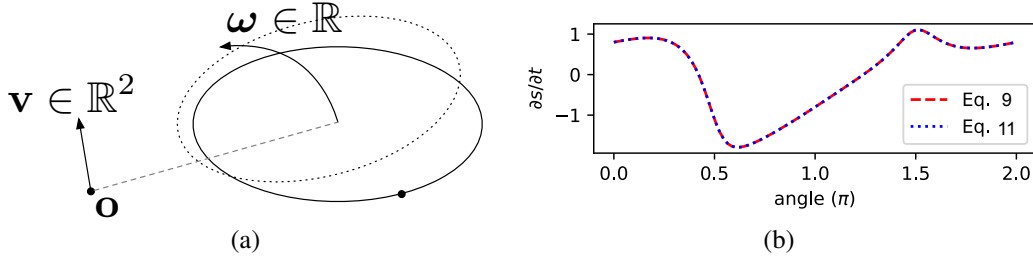

(a)                                              (b)

Figure 4: (a) 2D toy example of an ellipse moving with angular velocity $\omega$ and velocity $\boldsymbol{v}$. $\boldsymbol{o}$ is the origin. (b) We plot the SDF flow computed from Equation 9 and 11. Here, we use the polar coordinate system to represent points on the ellipse. The $x$-axis represents the points on the ellipse of different polar angles. The SDF flow computed from the scene flow (Equation 11) well matches that from the definition (Equation 9) for any point on the ellipse.

where $\boldsymbol{\omega} = [\frac{\partial \theta_x}{\partial t}, \frac{\partial \theta_y}{\partial t}, \frac{\partial \theta_z}{\partial t}]^T$ is the angular velocity of the surface ($\theta_x, \theta_y, \theta_z$ are the 3 rotation angles); $\boldsymbol{v} = \frac{\partial \mathbf{T}}{\partial t} \in \mathbb{R}^3$ is the velocity. The angular velocity and velocity define the 3D surface motion and thus the scene flow. The detailed derivation is provided in Section A.2. As will be demonstrated in Section 4, we would also like to compute scene flow directly from SDF flow with the derived relation linking them. To this end, we first transform Equation 11 to

$$\frac{\partial s}{\partial t} = -\boldsymbol{a}^T \begin{bmatrix} \boldsymbol{\omega} \\ \boldsymbol{v} \end{bmatrix} , \tag{12}$$

where $\begin{bmatrix} \boldsymbol{\omega} \\ \boldsymbol{v} \end{bmatrix} \in \mathbb{R}^6$, $\boldsymbol{a}_{1:3} = \boldsymbol{x} \times \boldsymbol{n}(\boldsymbol{x})$ and $\boldsymbol{a}_{4:6} = \boldsymbol{n}(\boldsymbol{x})$. In principle, given the SDF flow of at least 6 points that are moving rigidly, one can solve for the scene flow [1]. In practice, to handle the noise, we select more than 6 points and obtain the optimal scene flow by minimizing the least-square error (details are in Section A.8).

**Toy example.** In Figure 4(a), we provide a 2D toy example to verify Equation 11, where the initial position of an ellipse as well as its angular velocity $\omega \in \mathbb{R}$ and velocity $\boldsymbol{v} \in \mathbb{R}^2$ are given. As shown in Figure 4(b), the SDF flow computed from the scene flow (Equation 11) closely matches that from the definition (Equation 9).

## 4 EXPERIMENTS

### 4.1 DATASETS

We evaluate our method quantitativly on the CMU Panoptic dataset (Joo et al., 2017) and qualitatively on the Tensor4D dataset (Shao et al., 2023).

**The CMU Panoptic dataset** (Joo et al., 2017) captures various kinds of scenes including multi-person activities and human object interactions using multiple RGB(-D) cameras. Each scene is

---

[1]Note that, there exists exceptions where the solution may not match the true scene motion. We discuss such exceptions in Section A.3

Table 1: **Quantitative results on the CMU Panoptic dataset.** We report the accuracy (top), completeness (middle), and overall (bottom) in millimeter. For each sequence, the accuracy and completeness are averaged across all 24 frames, and the "avg" column is the average over all 5 scenes.

| acc (mm) | Ian3 | Haggling_b2 | Band1 | Pizza1 | Cello1 | avg |
|---|---|---|---|---|---|---|
| NDR (Cai et al., 2022) | 21.8 | 12.5 | 15.9 | 17.7 | 23.1 | 18.2 |
| Tensor4D (Shao et al., 2023) | 15.4 | 13.7 | 17.1 | 18.3 | 17.9 | 16.5 |
| Ours | **14.1** | **8.3** | **13.0** | **11.5** | **12.3** | **11.8** |
| comp (mm) | Ian3 | Haggling_b2 | Band1 | Pizza1 | Cello1 | avg |
| NDR (Cai et al., 2022) | 20.7 | 22.8 | 23.7 | 25.0 | 19.5 | 22.3 |
| Tensor4D (Shao et al., 2023) | 22.8 | 25.3 | 29.2 | 27.4 | 23.5 | 25.6 |
| Ours | **17.5** | **18.6** | **21.4** | **20.6** | **15.2** | **18.7** |
| overall (mm) | Ian3 | Haggling_b2 | Band1 | Pizza1 | Cello1 | avg |
| NDR (Cai et al., 2022) | 21.3 | 17.7 | 19.8 | 21.3 | 21.3 | 20.3 |
| Tensor4D (Shao et al., 2023) | 19.1 | 19.5 | 23.2 | 22.9 | 20.7 | 21.1 |
| Ours | **15.8** | **13.5** | **17.2** | **16.1** | **13.7** | **15.2** |

captured by 10 RGB-D and hundreds of RGB cameras. In this paper, we only use the images from 10 RGB-D cameras. We obtain the ground-truth point cloud at each time step by registering the depth maps taken from those cameras using the provided camera poses and intrinsics. We select 5 challenging clips: "Ian3", "Haggling_b2", "Band1", "Pizza1", and "Cello1". Our selected sequences cover activities like multi-person socializing ("Haggling_b2"), a band with multiple persons and musical instruments ("Band1") and a mother playing with a little child ("Ian3"). Each clip contains 24 frames from 10 camera view thus 240 images. The resolution of the image is $1920 \times 1080$. Since our goal is 3D reconstruction, we use all 10 camera views for training and only evaluate the meshes.

**The Tensor4D dataset** (Shao et al., 2023) is captured by a sparse-view system with RGB cameras. It contains a single person performing different actions like thumbs-up and waving hands. We select the 3 sample sequences provided on their official github page: "Boxing_v12", "Dance_v4", and "Thumbsup_v4". The "Boxing_v12" sequence is captured by 12 cameras in a circle surrounding the human. The "Dance_v4", and "Thumbsup_v4" sequences are taken by 4 forward-facing cameras. For each sequence, we select a clip of 12 frames. The image resulution is $1024 \times 1024$. Since no ground-truth geometry is available, we only provide qualitative comparison on this dataset.

### 4.2 METRICS, BASELINES & IMPLEMENTATION

**Metrics.** We follow the standard evaluation protocol in the multi-view stereo literature (Yao et al., 2018) to evaluate our method with accuracy, completeness and overall distance. Specifically, given the ground-truth point cloud $\bar{\mathcal{P}}$, and the predicted point cloud $\mathcal{P}$, the accuracy and completeness are defined as

$$\text{Acc} = \frac{1}{|\mathcal{P}|} \sum_{\boldsymbol{p} \in \mathcal{P}} \min_{\bar{\boldsymbol{p}} \in \bar{\mathcal{P}}} \|\boldsymbol{p} - \bar{\boldsymbol{p}}\|_2 \tag{13}$$

$$\text{Comp} = \frac{1}{|\bar{\mathcal{P}}|} \sum_{\bar{\boldsymbol{p}} \in \bar{\mathcal{P}}} \min_{\boldsymbol{p} \in \mathcal{P}} \|\boldsymbol{p} - \bar{\boldsymbol{p}}\|_2 . \tag{14}$$

The overall distance is the average of the accuracy and completeness.

**Baselines.** We compare our method with two recent NeRF-based dynamic scene reconstruction methods: NDR (Cai et al., 2022) and Tensor4D (Shao et al., 2023). NDR (Cai et al., 2022) attempts to find a bijective mapping between the observation space and the canonical space. Tensor4D (Shao et al., 2023) decomposes the 4D space into several 2D planes to speed up the model training. For both methods, we use their official implementation.

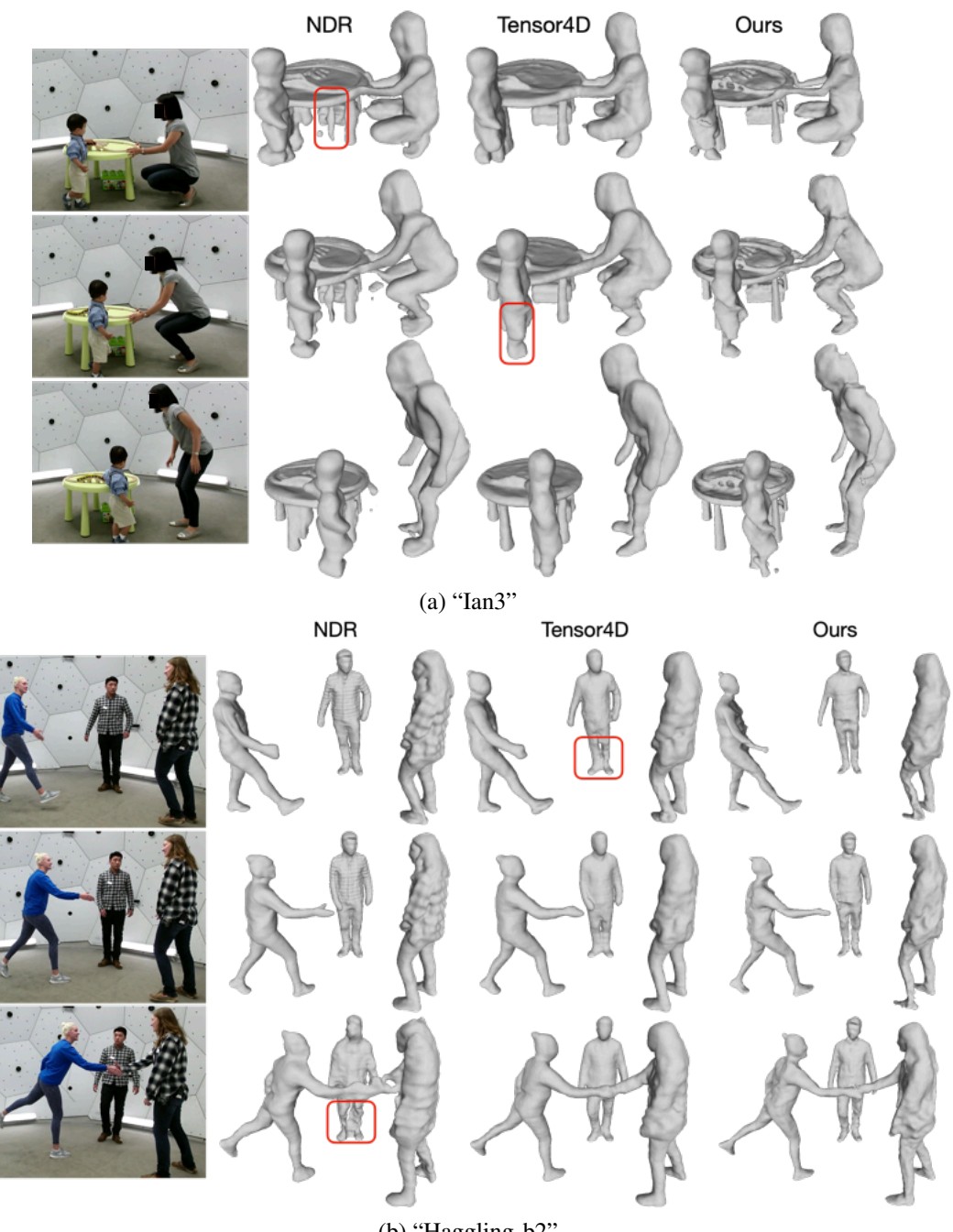

(a) "Ian3"

(b) "Haggling_b2"

Figure 5: **Qualitative results on the CMU Panoptic dataset.** Best viewed on screen.

**Implementation details.** We implement our method using Pytorch (Paszke et al., 2017) and use the Adam optimizer (Kingma & Ba, 2014) to train our model with a 0.0005 learning rate. The batch size is set to 1024. We use the second-order Runge-Kutta method to solve the integration in Equation 5. We train our model for 2000 epochs, which takes around 7 days on 2 NVIDIA 4090 GPUs for ten $1920 \times 1080$ videos of 24 frames. The rendering of one ray takes around 1.5 ms. The balancing weight $\lambda$ is set to 0.1. During testing, for all baselines and our method, we construct a 3D grid of resolution $512 \times 512 \times 512$ and query the SDF of each voxel in this grid from the trained model. We then use the marching cube algorithm (Lorensen & Cline, 1998) to obtain the mesh. We uniformly sample 10000 points from the 3D mesh and compare them to the ground-truth point clouds.

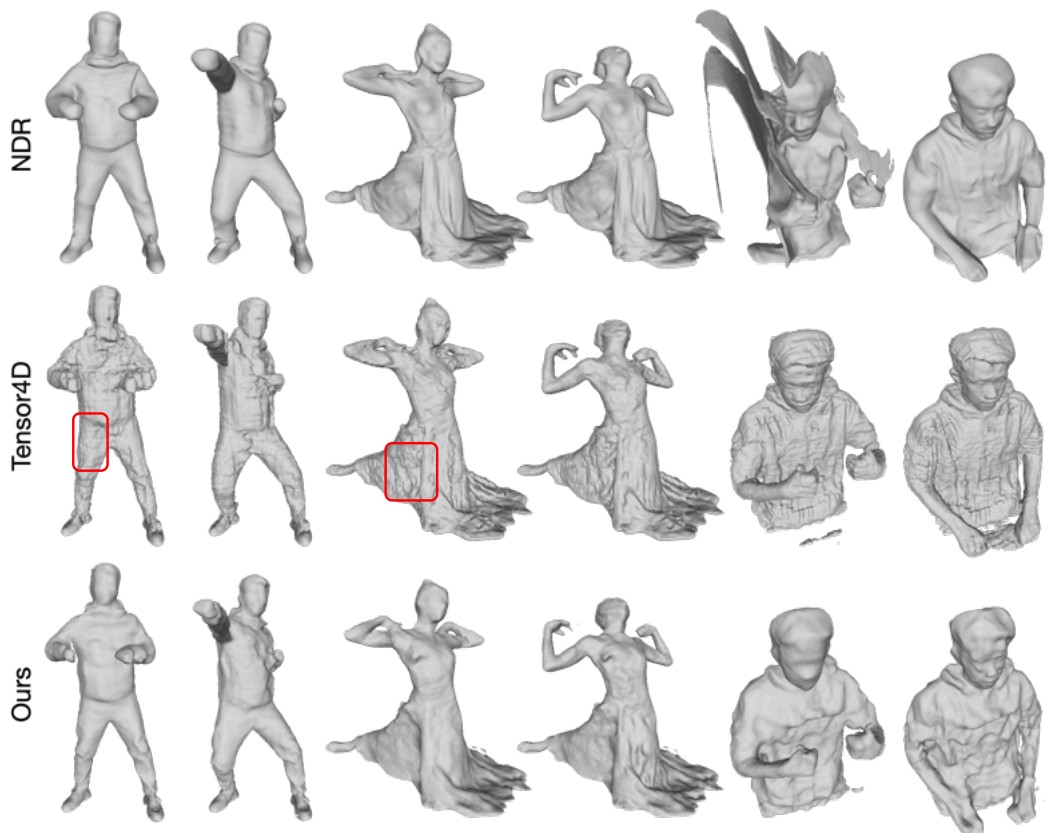

Figure 6: **Qualitative results on 3 samples of the Tensor4D dataset.**

### 4.3 RESULTS

**Quantitative results.** We provide quantitative results on the CMU Panoptic dataset (Joo et al., 2017) in Table 1. Our method consistently outperforms the baselines for all scenes in all metrics. For each scene, the reported accuracy (top), completeness (middle), and overall (bottom) are averaged across all frames, and we also report the average distance over all scenes (last column).

**Qualitative comparisons on the CMU Panopic dataset.** We compare our results to those of the baselines on the CMU Panopic dataset in Figure 5. Here, we show the reconstruction results of 3 different time steps for 2 scenes: "Ian3" and "Haggling_b2". As highlighted by the red box, the baselines (second and third columns) sometimes reconstruct overly smooth surfaces or even produce non-existing geometry. By contrast, the reconstructed meshes from our method are sharper with more details. More qualitative results on this dataset are provided in Section A.4 and the supplementary video.

**Qualitative comparisons on the Tensor4D dataset.** The results are shown in Figure 6. Although NDR (Cai et al., 2022) performs well on "Dance_v4" (middle), it sometimes fails on "Thumbsup_v4" (right) and its reconstructions of "Boxing_v12" (left) are over smoothed. As for Tensor4D (Shao et al., 2023), although it can sometimes reconstruct more details, such as the face in 'Thumbsup_v4", all of its results share similar artifacts which we conjecture are due to the tensor decomposition. Our method performs comparably to the baselines, with fewer artifacts.

Table 2: **Optical flow evaluation on the CMU Panoptic dataset.** We report the average end-point-error (EPE) in pixel.

| EPE (pixel) | Ian3 | Haggling_b2 | Band1 | Pizza1 | Cello1 | avg |
|---|---|---|---|---|---|---|
| NDR (Cai et al., 2022) | 6.86 | 4.43 | 1.66 | 3.11 | 2.41 | 3.69 |
| Ours | **3.49** | **2.97** | **1.18** | **1.18** | **1.34** | **2.04** |

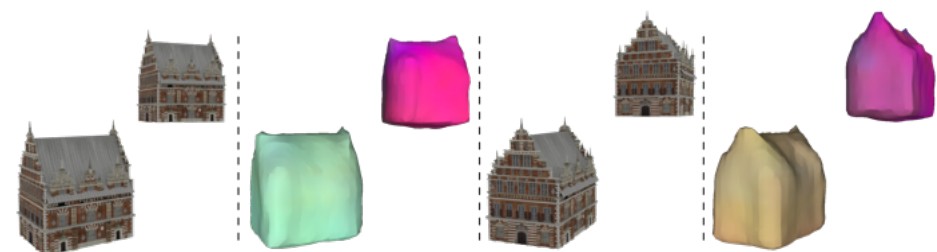

Figure 7: **Scene flow of a toy example.** We show the images at two different time steps (the first and third column) and their scene flow (the second and forth column). Different colors represent different scene flows.

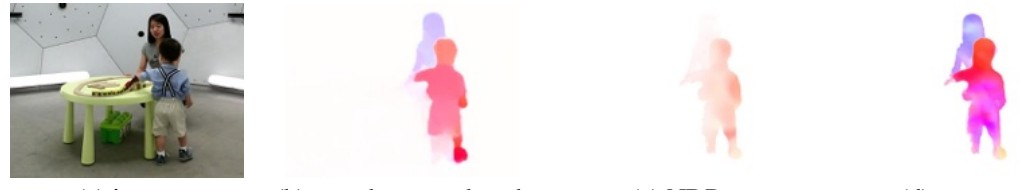

    (a) image        (b) pseudo-ground truth     (c) NDR       (d) ours

Figure 8: **The comparison of optical flow from NDR (Cai et al., 2022) and our SDF flow.** The optical flow computed using our SDF flow better matches the ground truth.

**Scene flow from SDF flow.** We further demonstrate that using our SDF flow lets us derive the scene flow, i.e., the angular velocity and velocity. We first show the derived scene flow of a toy example where two rigid objects are moving in the scene. For any surface point, we use the surrounding surface points to compute the scene flow. The results are shown in Figure 7. Note that, here, we do not have any prior knowledge of the number of rigid objects in the scene. As shown in the second and fourth column, the scene flow clearly distinguishes the two moving objects. We provide quantitative evaluation on the scene flow of this synthetic sequence in Section A.9.

We also compare our scene flow to that of NDR (Cai et al., 2022) on the real-world sequences of the CMU Panopic dataset. Since the ground-truth scene flow is not available, we evaluate the optical flow instead and use the optical flow estimated from RAFT (Teed & Deng, 2020) as pseudo-ground truth. Specifically, we project our scene flow to the image plane to obtain the corresponding optical flow. For NDR (Cai et al., 2022), we use the bijective mapping to obtain the scene flow. We report the average end-point-error of the optical flow. The evaluation results is shown in Table 2. As also evidenced in Figure 8, our optical flow better matches the ground truth. We also visualize the 3D scene flow in Section A.5 and the supplementary video. Our scene flow better reflects the real scene motion.

## 5   CONLUSION

In this paper, we have proposed to exploit the SDF flow to represent a dynamic scene with the first-order derivative of its SDF with respect to time. Our SDF flow naturally captures the deformations of the scene surface. We have designed a NeRF-based pipeline using our SDF flow to reconstruct a dynamic scene from multi-view videos. Our experiments show that our method yields state-of-the-art performance. We have further derived a mathematical relation between the SDF flow and the scene flow. Such a relation allows us to calculate the scene flow from the SDF flow analytically by solving linear equations. We have demonstrated that the resulting scene flow correctly reflects the real motion. In the future, we would like to explore the potential of applying our method to monocular videos.

## ACKNOWLEDGEMENTS

This research was supported in part by the Australia Research Council DECRA Fellowship (DE180100628) and ARC Discovery Grant (DP200102274). The authors would like to thank NVIDIA for the donated GPU (Titan V).

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

## A  APPENDIX

### A.1  PROOF OF THEOREM 2

*Proof.* Let us first prove the absolute value of SDF flow,

$$|\frac{\partial s}{\partial t}| = \lim_{\Delta t \to 0} |\frac{\Delta s}{\Delta t}| \tag{15}$$

$$= \lim_{\Delta t \to 0} \frac{\|\boldsymbol{x}'' - \boldsymbol{x}\|_2}{\Delta t} , \tag{16}$$

where $\boldsymbol{x}''$ is the closest point on the evolved surface after time $\Delta t$ ($\Delta t > 0$).

Since the surface is locally smooth around $\boldsymbol{x}$ and its deformation is also smooth, as $\Delta t$ approaches zeros, $\boldsymbol{x}''$ is infinitely close to the corresponding point (denoted as $\boldsymbol{x}'$) of $\boldsymbol{x}$ on the evolved surface (as shown in Figure 3). Thus, we can regard $\boldsymbol{x}''$ as on the tangent plane to the evolved surface at $\boldsymbol{x}'$.

$$\lim_{\Delta t \to 0} \frac{\|\boldsymbol{x}'' - \boldsymbol{x}\|_2}{\Delta t} = \lim_{\Delta t \to 0} \frac{|(\boldsymbol{x}' - \boldsymbol{x})^T \boldsymbol{n}(\boldsymbol{x}')|}{\Delta t} \tag{17}$$

$$= |\frac{\partial \boldsymbol{x}}{\partial t}^T \boldsymbol{n}(\boldsymbol{x})| , \tag{18}$$

where $\boldsymbol{n}(\boldsymbol{x}')$ is the normal of $\boldsymbol{x}'$ on the evolved surface and $\lim_{\Delta t \to 0} \boldsymbol{n}(\boldsymbol{x}') = \boldsymbol{n}(\boldsymbol{x})$.

As to the sign of SDF flow, it is easy to see that if the angle between $\boldsymbol{x}' - \boldsymbol{x}$ and $\boldsymbol{n}(\boldsymbol{x}')$ is obtuse, the SDF change is positive, otherwise it is negative.

We provide another form of the Theorem 2 wishing to help the readers to better understand it.

**Theorem 3** *Let $\mathcal{V}$ be a smooth surface in $\mathbb{R}^3$ with $\boldsymbol{x}$ a point on the surface and $\boldsymbol{n}(\boldsymbol{x})$ the normal to the surface at $\boldsymbol{x}$. Let $\boldsymbol{\gamma}(t)$ be a smooth curve $\boldsymbol{\gamma} : \mathbb{R}^{\geq 0} \to \mathbb{R}^3$ be a smooth path in $\mathbb{R}^3$ such that $\boldsymbol{\gamma}(0) = \boldsymbol{x}$. Let $s(t)$ be the signed distance from the curve to the surface. Then*

$$s'(0) = \boldsymbol{\gamma}'(t)^T \boldsymbol{n}(\boldsymbol{x}) . \tag{19}$$

This theorem is effectively saying that at time $t = 0$, the point $\boldsymbol{\gamma}(t)$ is moving away from the surface at a speed equal to the projection of the velocity $\boldsymbol{\gamma}'(0)$ onto the normal direction $\boldsymbol{n}(\boldsymbol{x})$. The proof can be made rigorous by invoking the implicit function theorem, which states that in chosen local coordinates, the surface can be expressed as a graph, locally around $\boldsymbol{x}$.

## A.2 DETAILED DERIVATION OF THE RELATION BETWEEN SCENE FLOW AND SDF FLOW

Let us start from Equation 8, when $\Delta t$ approaches zero, the rotation angles are very small. According to Rodrigues' rotation formula, we can approximate the rotation as

$$\Delta \mathbf{R} \approx \mathbf{I} + \Delta \mathbf{K} , \tag{20}$$

where $\Delta \mathbf{K} = \begin{bmatrix} 0 & -\Delta \theta_z & \Delta \theta_y \\ \Delta \theta_z & 0 & -\Delta \theta_x \\ -\Delta \theta_y & \Delta \theta_x & 0 \end{bmatrix}$ ($\Delta \theta_x, \Delta \theta_y, \Delta \theta_z$ are the Euler angles). We can then compute the scene flow as

$$\frac{\partial \boldsymbol{x}}{\partial t} = \lim_{\Delta t \to 0} \frac{\boldsymbol{x}' - \boldsymbol{x}}{\Delta t} \tag{21}$$

$$= \lim_{\Delta t \to 0} \frac{\Delta \mathbf{K} \boldsymbol{x} + \Delta \mathbf{T}}{\Delta t} \tag{22}$$

$$= \boldsymbol{\omega} \times \boldsymbol{x} + \boldsymbol{v} , \tag{23}$$

where $\boldsymbol{\omega} \in \mathbb{R}^3$ is the angular velocity and $\boldsymbol{v} \in \mathbb{R}^3$ is the velocity. Substituting the above equation into Equation 10, we have

$$\frac{\partial s}{\partial t} = -(\boldsymbol{\omega} \times \boldsymbol{x} + \boldsymbol{v})^T \boldsymbol{n}(\boldsymbol{x}) . \tag{24}$$

## A.3 EXCEPTION

Given the SDF flow of 6 points on a surface that moves smoothly and rigidly, one can obtain the scene flow $\begin{bmatrix} \boldsymbol{\omega} \\ \boldsymbol{v} \end{bmatrix}$ of the surface by solving a linear equation

$$\boldsymbol{d} = -\mathbf{A} \begin{bmatrix} \boldsymbol{\omega} \\ \boldsymbol{v} \end{bmatrix} , \tag{25}$$

where $d \in \mathbb{R}^6$ is the SDF flow of those 6 points. $\mathbf{A} \in \mathbb{R}^{6 \times 6}$ whose rows are computed from the normals and locations of those 6 points. To the best of our knowledge, the only exception when we cannot obtain the correct scene flow is that when the scene motion does not lead to any SDF changes to any 6 points on the surface, thus, $d = \mathbf{0}$. In this case, there are many solutions to the linear equation. For example, a plane moving along any of its tangent direction or a ball rotating around its centre. For the former, any velocity on the tangent plane could be the solution. For the latter, any angular velocity by the ball centre will work. Note that, the linear relation between scene flow and SDF flow defined in Equation 11 still holds on these exceptions. In fact, as long as the scene is piece-wise rigid and scene motion is smooth, Equation 11 holds.

### A.4 Additional Qualitative Results on the CMU Panoptic dataset

We provide qualitative results on the rest 3 sequences of the CMU Panoptic dataset in Figure 9,10 and 11 below.

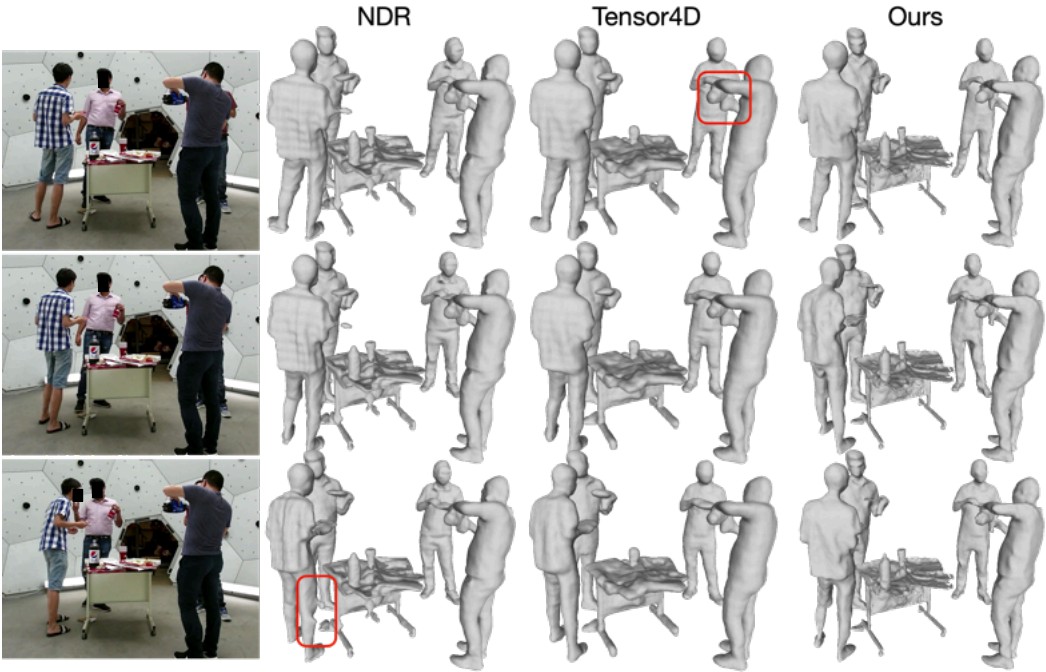

Figure 9: **Qualitative results on "Pizza1" sequence of the CMU Panoptic dataset.**

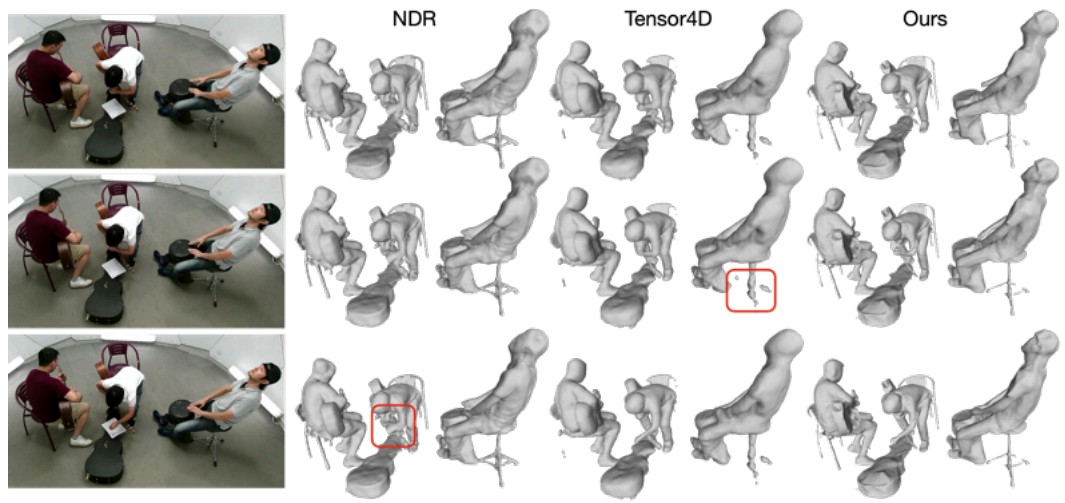

Figure 10: **Qualitative results on "Band1" sequence of the CMU Panoptic dataset.**

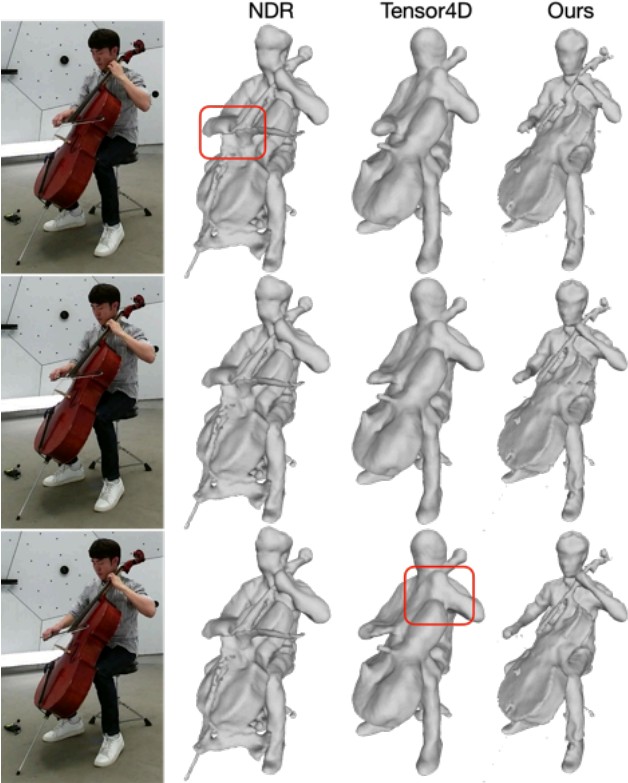

Figure 11: **Qualitative results on "Cello1" sequence of the CMU Panoptic dataset.**

### A.5 MORE QUALITATIVE RESULTS ON SCENE FLOW

We visualize the scene flow for all 5 sequences of the CMU Panoptic dataset in Figure 12, 13, 14, 15 and 16 below.

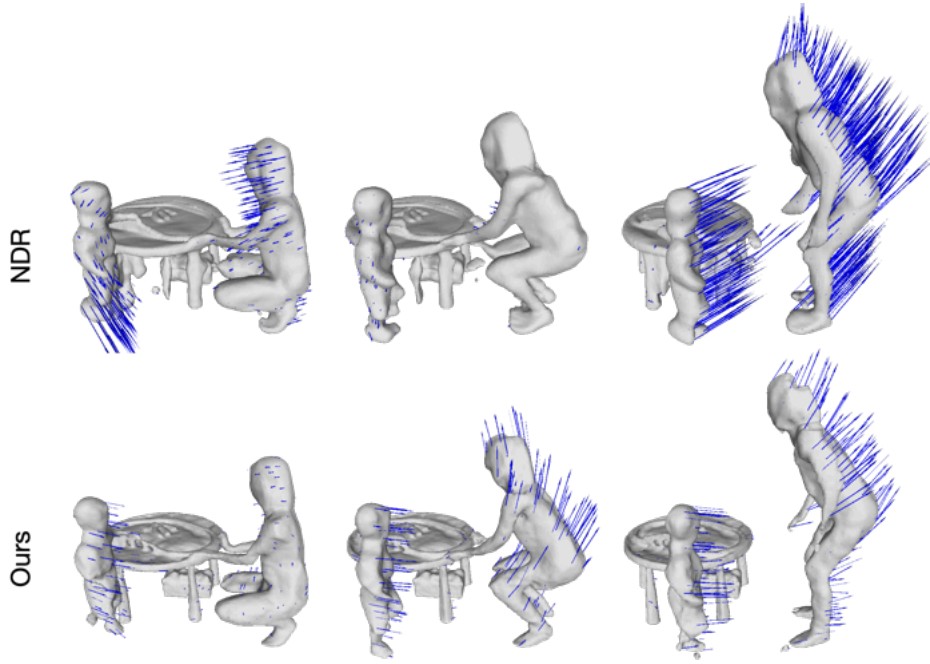

Figure 12: **Scene flow of the "Ian3" sequence.** We show the reconstructed meshes as well as the scene flow (blue arrows) at different time steps. Note that, the blue arrows are scaled for better visualization. The scene flow from NDR are wrong while the scene flow from our SDF flow correctly reflects the true 3D motion.

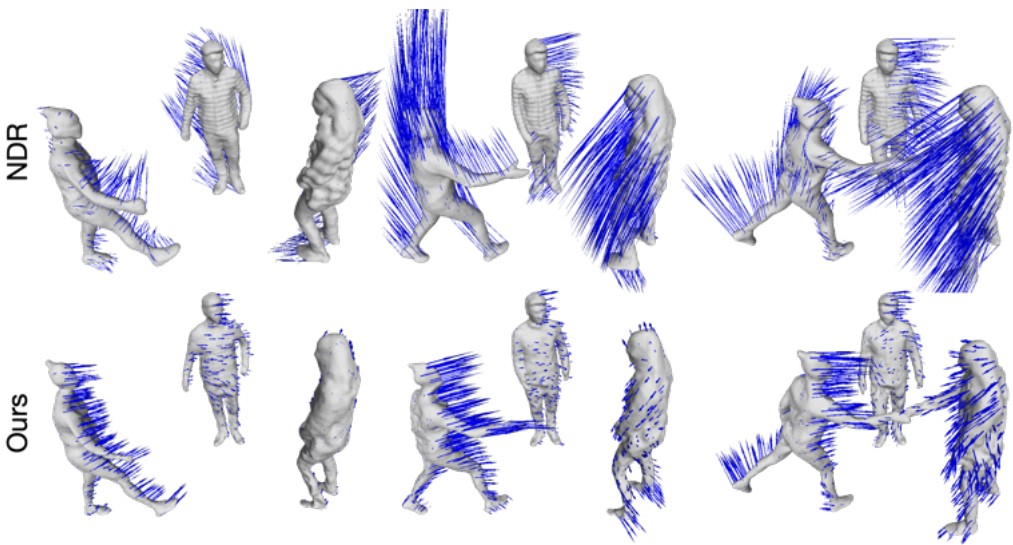

Figure 13: **Scene flow on "Hanggling_b2" of the CMU Panoptic dataset.**

### A.6 MULTI-VIEW IMAGES

In Figure 17, we show the 10 images from different camera views of "Ian3". The cameras are fixed for all the 5 sequences of the CMU Panoptic dataset cross all time steps. It is a challenging setup due to the large view changes.

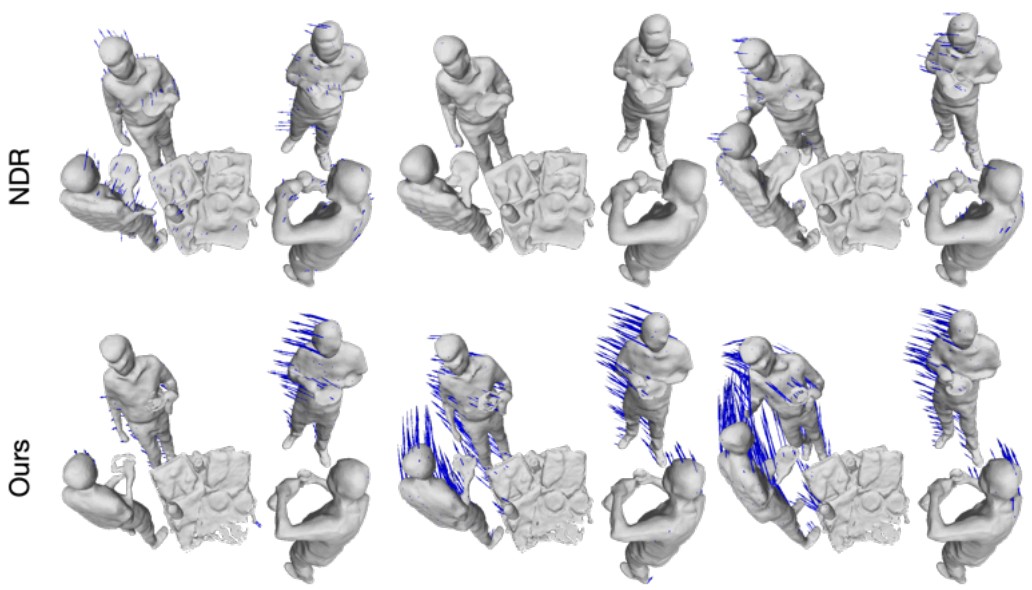

Figure 14: **Scene flow on "Pizza1" of the CMU Panoptic dataset.**

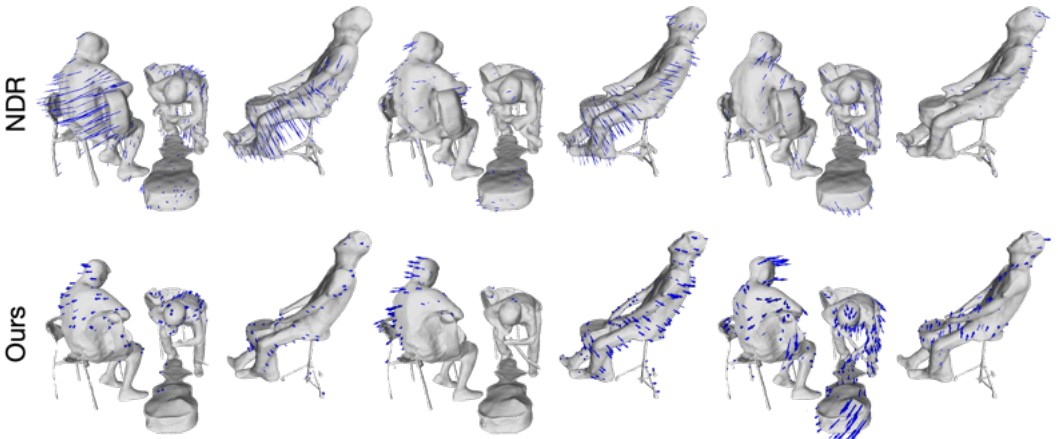

Figure 15: **Scene flow on "Band1" of the CMU Panoptic dataset.**

## A.7    SCENE FLOW REGULARIZATION

We believe that revealing the linear relationship between the SDF flow and scene flow will be valuable for modeling dynamic scenes. We evidence this by the toy example shown in Figure 7. In particular, we train a new model (named "Ours w/ reg") that outputs not only the SDF flow but also the scene flow

$$\frac{\partial s(\boldsymbol{x},t)}{\partial t}, \boldsymbol{\omega}(\boldsymbol{x},t), \boldsymbol{v}(\boldsymbol{x},t) = f(\boldsymbol{x},t) \ . \tag{26}$$

Given the estimated SDF flow and scene flow, we can then apply several priors.

**SDF flow and scene flow consistency prior.** According to Equation 11, for any point $\boldsymbol{x}$ on the surface, we have

$$\mathcal{L}_{\text{SF}} = \|(\boldsymbol{\omega}(\boldsymbol{x},t) \times \boldsymbol{x} + \boldsymbol{v}(\boldsymbol{x},t))^T \boldsymbol{n}(\boldsymbol{x},t) + \frac{\partial s(\boldsymbol{x},t)}{\partial t}\|_2^2 \ . \tag{27}$$

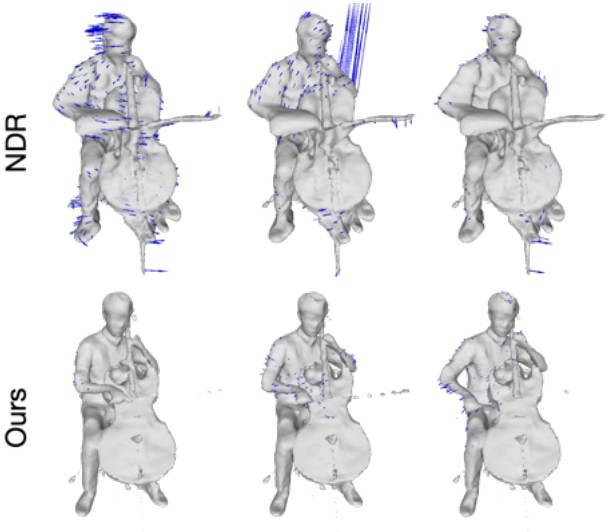

Figure 16: **Scene flow on "Cello1" of the CMU Panoptic dataset.**

**Piece-wise rigid prior.** We also encourage the scene flow of neighbouring points to be similar.

$$\mathcal{L}_{\text{Rigid}} = \sum_{\boldsymbol{y} \in \mathcal{N}_\epsilon(\boldsymbol{x})} |\boldsymbol{\omega}(\boldsymbol{x}, t) - \boldsymbol{\omega}(\boldsymbol{y}, t)| + |\boldsymbol{v}(\boldsymbol{x}, t) - \boldsymbol{v}(\boldsymbol{y}, t)| \tag{28}$$

**Optical flow loss.** For any point $\boldsymbol{x}$ at frame $t_1$, we can obtain its corresponding point at next frame $t_2$ as

$$\boldsymbol{x}'(\boldsymbol{x}) = \int_{t_1}^{t_2} \boldsymbol{\omega}(\boldsymbol{x}, t) \times \boldsymbol{x} + \boldsymbol{v}(\boldsymbol{x}, t) dt. \tag{29}$$

Given the 3D correspondences, we can use the camera parameters and the standard volume rendering techniques to compute the optical flow for any pixel $\boldsymbol{r}$ on the image.

$$\boldsymbol{f}(\boldsymbol{r}) = \int_{r_n}^{r_f} T(r)\sigma(\boldsymbol{r}(r))(\Pi(\boldsymbol{x}'(\boldsymbol{x})) - \Pi(\boldsymbol{x})) dr \;, \tag{30}$$

where $\Pi(\cdot)$ is the prospective camera project. The optical flow loss is then defined as

$$\mathcal{L}_{\text{OF}} = \|\boldsymbol{f}(\boldsymbol{r}) - \bar{\boldsymbol{f}}(\boldsymbol{r})\|_2^2 \;, \tag{31}$$

where $\bar{\boldsymbol{f}}(\boldsymbol{r})$ is the ground truth optical flow.

The final loss is the weighted sum of all losses

$$\mathcal{L} = \mathcal{L}_{\text{RGB}} + \lambda \mathcal{L}_{\text{SDF}} + \lambda_{\text{SF}} \mathcal{L}_{\text{SF}} + \lambda_{\text{Rigid}} \mathcal{L}_{\text{Rigid}} + \lambda_{\text{OF}} \mathcal{L}_{\text{OF}} \;. \tag{32}$$

We report the reconstruction results in Table 3. The scene flow regularization helps to improve the reconstruction quality.

### A.8 SCENE FLOW FROM SDF FLOW

Given a set of points $\mathbb{P}$ sampled uniformly from a reconstructed mesh, for any point $\boldsymbol{x}$ in this set, we first obtain its $K$ nearest neighbours $\{\boldsymbol{y}_i\}_{i=1}^K$, where $\boldsymbol{y}_i \in \mathbb{P}$. For every neighbouring points $\boldsymbol{y}_i$, we query its surface normal $\boldsymbol{n}(\boldsymbol{y}_i)$ and SDF flow $d(\boldsymbol{y}_i)$ from the MLPs. According to Equation 17, we obtain a matrix $\mathbf{A} \in \mathbb{R}^{K \times 6}$ where the $i$-th row $\mathbf{A}_i = [(\boldsymbol{y}_i \times \boldsymbol{n}(\boldsymbol{y}_i))^T, \boldsymbol{n}(\boldsymbol{y}_i)^T]$ and the SDF flow vector $\boldsymbol{d} \in \mathbb{R}^K$ where $\boldsymbol{d}_i = d(\boldsymbol{y}_i)$. We can then solve such problem as

$$\begin{bmatrix} \boldsymbol{\omega}^* \\ \boldsymbol{v}^* \end{bmatrix} = \arg\min_{\boldsymbol{\omega},\boldsymbol{v}} \|\boldsymbol{d} + \mathbf{A} \begin{bmatrix} \boldsymbol{\omega} \\ \boldsymbol{v} \end{bmatrix}\|_2^2 \tag{33}$$

$$= -(\mathbf{A}^T \mathbf{A})^{-1} \mathbf{A}^T \boldsymbol{d} \tag{34}$$

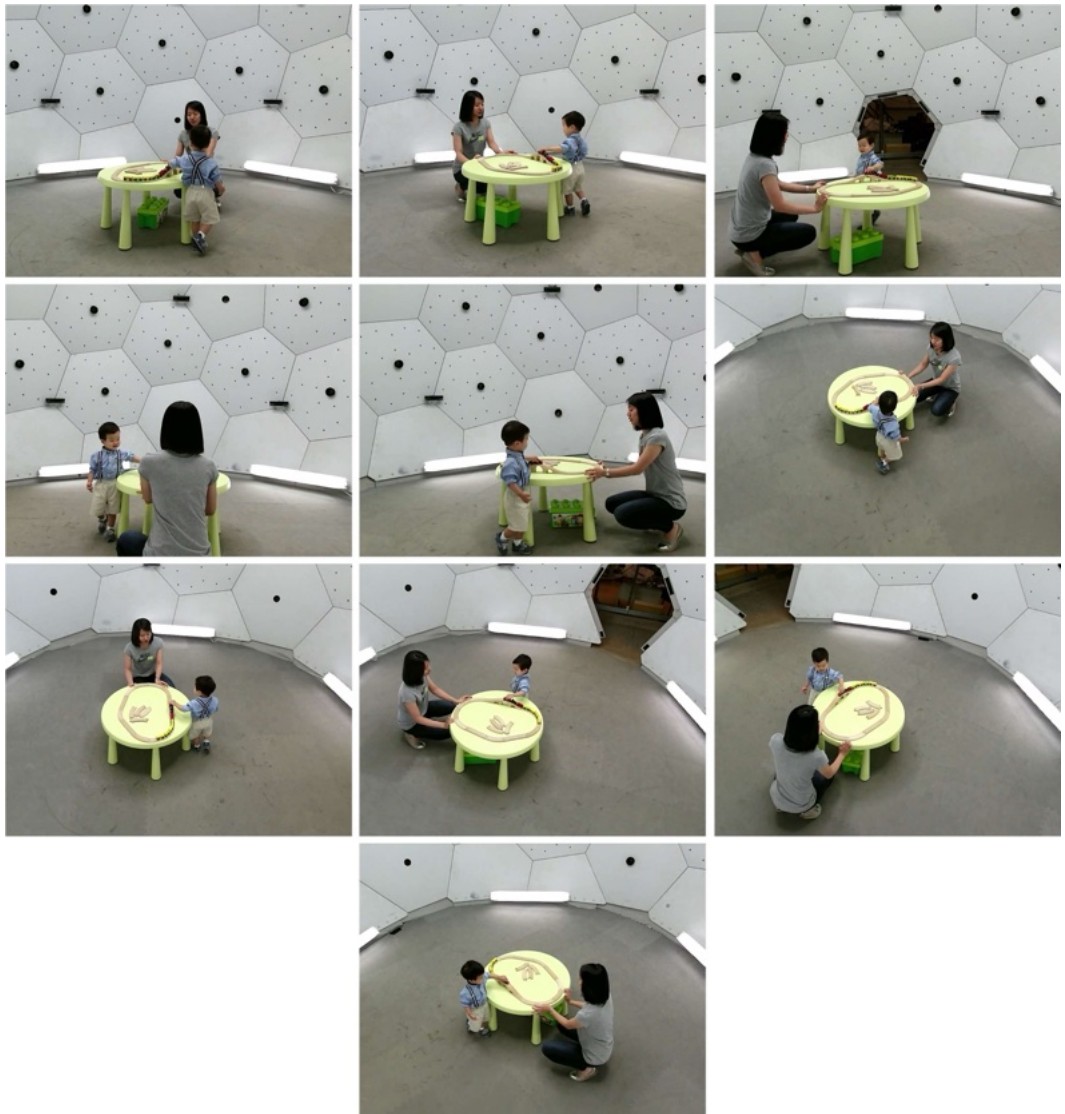

Figure 17: **Multi-view images used in our experiments.** It is a challenging setup due to the small overlap between different views. All 5 sequences of the CMU Panoptic dataset use the same camera setup. The images are zoomed and cropped to better show the content.

where $\boldsymbol{\omega}^*$, $\boldsymbol{v}^*$ is the optimal angular velocity and velocity.

In practice, we uniformly sample around 2000 points from a reconstructed mesh and choose around 200 neighbouring points to compute the scene flow of one surface point. To further study the influence of the number of neighbouring point, we compare the scene flow end-point-error(EPE in $mm$) with respect to the number of neighbouring points in Table 4 on the synthetic sequence.

## A.9 SCENE FLOW EVALUATION

To directly evaluate the scene flow, we use the synthetic sequence shown in Figure 7 where we have the ground truth scene flow across all frames. The evaluation results are shown in Table 6. We compare the results of three models: NDR (Cai et al., 2022), Ours, Ours with regularization which we will introduce in the next section. Our model without any explicitly scene flow regularization performs better than NDR (Cai et al., 2022). With further regularization, the scene flow can be drastically improved. We also evaluate the long-term correspondences up to 5 frames. In particular,

Table 3: **Reconstruction results on the synthetic sequence.** We compare the results from NDR (Cai et al., 2022), our model with only the SDF flow ("Ours"), and with scene flow regularization ("Ours w/ reg").

|  | acc | comp | overall |
|---|---|---|---|
| NDR (Cai et al., 2022) | 33.5 | 25.3 | 29.4 |
| Ours | 31.8 | 23.3 | 27.6 |
| Ours w/ reg | **27.2** | **19.1** | **23.2** |

Table 4: **Scene flow error v.s. number of neighbouring points.**

| $K$ | 50 | 100 | 200 | 500 | 1000 |
|---|---|---|---|---|---|
| EPE (mm) | 13.7 | 12.6 | 11.7 | 11.3 | 12.0 |

for every surface point in a frame, we compute the scene flow not only to the next frame but also to further future frames. The scene flow error increases with respect to frame gaps.

Table 5: **Scene flow evaluation on the synthetic data.** We report the average end-point-error (EPE) in millimeter. We evaluate the long-term correspondences up to 5 frames. In particular, for every surface point in a frame, we compute the scene flow to the future 1, 2, and 5 frames.

| EPE (mm) | +1 | +2 | +5 |
|---|---|---|---|
| NDR (Cai et al., 2022) | 16.8 | 34.8 | 90.2 |
| Ours | 11.7 | 24.7 | 71.4 |
| Ours w/ reg | **5.8** | **13.0** | **51.9** |

## A.10 LIMITATIONS

Similar to all other multi-view reconstruction works, the reconstruction quality highly rely on the number of views. Another limitation of our SDF flow is the time complexity. To solve for the integral (Equation 5) for any query point, the number of function evaluation grows with respect to time. We aim to address those limitations in the future work. For the first limitation, since we now have an explicit relation between the scene geometry and scene motion, we may be able to achieve better geometry with extra regularization from motion such as optical flow. For the second limitation, a potential solution could be that instead of integrating from the initial time step, we can integrate within a fixed time window.

## A.11 SDF V.S. SDF FLOW

We compare the reconstruction results of our model directly output the SDF and SDF flow. During training we set aside several frames to evaluate the interpolation ability of both representations. In particular, we train the model on $\{1, 4, 7, 10\}$-th frames and the evaluation results on those frames are refered to as reconstruction and those on $\{2, 3, 5, 6, 8, 9\}$-th frames are referred to as interpolation. Our model with SDF flow representation consistently outperforms that with SDF representation especially at the unseen frames (interpolation).

Table 6: **SDF v.s. SDF flow with the same model on the CMU Panoptic dataset.**

| | Reconstruction | | | Interpolation | | |
|---|---|---|---|---|---|---|
| | acc | comp | overall | acc | comp | overall |
| Ours w/ SDF | 12.4 | 19.0 | 15.7 | 13.0 | 20.8 | 16.9 |
| Ours w/ SDF flow | **11.6** | **18.9** | **15.2** | **11.5** | **18.9** | **15.2** |

