# OpenReview forum: "Neural SDF Flow for 3D Reconstruction of Dynamic Scenes"
_ICLR.cc/2024/Conference — ICLR 2024 poster_

### Official Review · Reviewer_uMrz · 2023-10-17

**Soundness:** 2 fair
**Presentation:** 3 good
**Contribution:** 3 good
**Rating:** 8
**Confidence:** 5

**Summary:**

The paper tackles dynamic surface reconstruction and scene-flow estimation from multi-view RGB video with known camera parameters. It uses a global coordinate-based MLP that outputs the SDF in canonical space (t=0), which then gets converted to density via VolSDF's formula, enabling NeRF-style rendering. Training uses an RGB reconstruction loss and an Eikonal regularizer for the SDF. The method departs from prior dynamic NeRFs in its deformation model: a separate MLP is trained to predict the temporal change (derivative) of the SDF at any point in space. These changes get integrated with Runge-Kutta over time to obtain the time-dependent SDF. Furthermore, the paper shows how this SDF flow can be converted into scene flow. Experiments show that the geometry is either less noisy or more detailed than prior methods on the CMU Panoptic dataset and another existing dataset.

**Strengths:**

Expanding the dynamic NeRF field in the direction of reconstruction/geometry is worthwhile in my opinion, as it enables more applications than novel view synthesis alone would. Correspondences are also interesting for the same reason. Thus, the problem setting is a big strength of the paper.

The method's SDF flow parametrization is novel, as far as I know. It seems generic enough to be potentially valuable for problem settings very different from the setting in the submission.

The illustrations are done well and the paper in general is written very well.

The method outperforms prior methods quantitatively. Qualitatively, it is better in at least one aspect compared to each prior method, while on par in the other aspects, and hence overall better.

**Weaknesses:**

*** Method:

[Tangential Motion] The paper should more clearly state that tangential motion causes issues when converting a time-dependent SDF into scene flow.

[Topology Change] The results don't show any good instances of topology change. I do not understand this claim. The best example I could find is at 0:45. Are there any other good instances? I'm also lost as to which aspect of the method helps with topology change. Please mark that more clearly.

*** Experiments:

[Scene Flow Evaluation] The paper doesn't show any appearance/novel view results. That's okay since it focuses on the geometry, and appearance is only used as an auxiliary to deal with the RGB input. However, I would still be curious to see video results where the appearance is taken from t=0 and propagated via the SDF flow. That would allow to visualize longer-term correspondence drift and whether tangential motion causes issues in practice. Currently, the SDF flow (and thus the derived scene flow) only cares about short-term "geometrical" correspondences. - This is relevant because the paper doesn't just claim that the SDF flow is a helpful parametrization but rather that it enables scene flow, i.e. correspondences. Currently, the qualitative and quantitative scene flow evaluation is only short-term, even though the method requires long-term correspondences (equation 7). The quantitative scene flow evaluation is also only a single sequence.

[Integral Evaluation] How does the temporal integration scale with scene length? Presumably linearly? Is that why only 24 frames are used? How many function evaluations does the Runge-Kutta solver need for the last (24th) frame?

[Weak Qualitative Geometry] Tensor4D gives detailed, noisy results and NDR is less noisy but lacks detail. The submission's results have the level of detail of Tensor4D and the low noise level of NDR. Still, none of these results are overwhelming or that impressive. (Doesn't need to be addressed in the rebuttal.)

*** Paper:

[Related Work] The related work section isn't that thorough. For example, the dynamic NeRF papers Fang et al. Fast dynamic radiance fields with time-aware neural voxels and Li et al. Neural 3D Video Synthesis are missing. Li et al., like NSFF, can also handle topology changes. Furthermore, shape-from-template methods aren't mentioned. Scene flow methods (e.g. Song et al. PREF: Predictability Regularized Neural Motion Fields) aren't discussed. Also, since NDR is compared to, mentioning some of the papers in the RGB-D line of work would be good, e.g. DynamicFusion, VolumeDeform, KillingFusion, OcclusionFusion. And TARS (Duggal et al. Topologically-aware deformation fields for single-view 3d reconstruction) seems relevant since it isn't restricted to blend skinning (a typo ("blender") in the submission).

[Limitations] Please discuss limitations.

*** Minor:

[Input Assumptions] Since the mathematical derivation matters here, please state what assumptions go into the paragraph between equation 5 and 6, which talks about the continuity and differentiability of a time-dependent SDF. In theory, there is nothing preventing an object from appearing out of nowhere, an SDF does not inherently have any restrictions on its temporal evolution (while it does have restrictions in space, namely the Eikonal equation). In real-world cases, geometry noise (say, due to the sensor or an imperfect reconstruction) pops randomly into existence and vanishes randomly over time, which leads to discontinuities w.r.t. the time parameter. Unless point x is meant to be a Lagrangian particle rather than an Eulerian grid coordinate? Figure 2 looks Eulerian though. --- Please state the assumptions that go into that paragraph.

[Wrong Argument] The argument at the end of Sec. 3.1 is that most dynamic NeRF methods have issues with topology changes. That's not the case for most dynamic NeRF methods that handle general objects since most condition the canonical model in some manner on time (e.g. HyperNeRF). An extreme case of that is Neural Scene Flow Fields. The argument only holds for D-NeRF-style methods like Nerfies or NR-NeRF. For the others, the better argument would be to say that they use density to parametrize geometry rather than SDF and hence the geometry tends to be very noisy and lack a clearly defined surface.

[Labelling] Figure 2 would benefit from labelling the two ellipses with their respective timesteps (t=0 and t=6?).

**Questions:**

I have listed my concerns in Weaknesses. The concerns under "Minor" don't need to be addressed in a rebuttal, except that clearing up [Input Assumptions] would help me. For all others, I'd appreciate a response. In particular, the most important concerns I have are [Topology Change] and [Scene Flow Evaluation]. If these are not addressed in a rebuttal in some form, I am against accepting the paper. I would still want the rebuttal to address the other major concerns to feel like I have a decent grasp of the submission.

Overall, I lean towards reject. Even though the results are not that impressive, the paper shows nice technical contributions. If the rebuttal addresses my two main concerns well, I could increase my score to acceptable.

=====

Post-rebuttal justification: The rebuttal addressed all concerns very well.

---

> ### Author Response · Authors · 2023-11-19
> **Responses to Reviewer uMrz (Part 1)**
>
> - **Tangential Motion**
>
> Thank you for the suggestion. We have updated the exception analysis in Section A2 of the main script as follow.
>
> Given the SDF flow of 6 points on a surface that moves smoothly and rigidly, one can obtain the scene flow $\begin{bmatrix}\\boldsymbol\\omega\\\\\\mathbf{v}\end{bmatrix}$ of the surface by solving a linear equation
> $$
> \\mathbf{d} = -\\mathbf{A}\begin{bmatrix}
>                 \\boldsymbol\\omega\\\\
>                 \\mathbf{v}
>             \end{bmatrix},
> $$
> where $\\mathbf{d}\\in\\mathbb{R}^6$ is the SDF flow of those 6 points. $\\mathbf{A}\\in\\mathbb{R}^{6\\times6}$ whose rows are computed from the normals and locations of those 6 points. To the best of our knowledge, the only exception when we cannot obtain the correct scene flow is that when the scene motion does not lead to any SDF changes to any 6 points on the surface, thus, $\\mathbf{d}=\\mathbf{0}$. In this case, there are many solutions to the linear equation. For example, a plane moving along its tangent direction or a ball rotating around its centre. For the former, any velocity on the tangent plane could be the solution. For the latter, any angular velocity by the ball centre will work. Note that, the linear relation between scene flow and SDF flow defined in Equation 16 still holds on these exceptions. In fact, as long as the surface and scene motion is smooth, Equation 16 holds.
>
> We would also like to clarify that tangential motion does not necessarily lead to degenerate case when using the above linear equation to solve for scene flow. For example, given a ball translating in the space, there are always points on the ball where the velocity direction is along their tangent planes. However, we can always find another 2 points, together with any of those points, to solve for the velocity of the ball (we use 3 points because there is only translation.).
>
> - **Topology Change**
>
> There are a few topology change instances in the supplementary video: at 00:10 when the woman's hands leaving the table, at 00:19 when the two girls shaking hands, at 00:24, when the two hands of the guitarist in the middle are put apart.
>
> We acknowledge that all works based on SDF, such as Shao et al. (2023), including ours can handle the topology change. For those works that map observation frames to canonical frame such as Cai et al. (2022); Park et al. (2021a), they require to bring the input space into a higher dimension space to handle the topology change as has done in HyperNeRF (Park et al., 2021b). For other works that estimate per-frame densities and use scene flow to enforce the temporal constraints of the scene such as Li et al. (2021); Song et al. (2022), they can also handle the topology change due to the per-frame density estimation.
>
> At last we would like to emphasize that handling topology change is not the focus nor the contribution of our work. Our main contribution is the novel dynamic scene representation: SDF flow which can do whatever SDF-based representations can and we further derive the mathematical relation between SDF flow and scene flow which bridges the scene geometry with scene motion. We believe such relation will benefit the community of dynamic scene reconstruction. To avoid misunderstanding, we have revised paragraphs with topology changes

---

> ### Author Response · Authors · 2023-11-19
> **Responses to Reviewer uMrz (Part 2)**
>
> - **Scene Flow Evaluation**
>
> **Render results.** Thank you for the suggestion. We provide the meshes with texture and rendered videos in the updated supplementary material (named *supp\_vid\_rebuttal.mp4* and *textured_meshes/*). The color of a surface point is obtained either from the current time step or from the corresponding point in future time steps (specified by the title of the file. For example, for the mesh file named *geo_1_col_2.ply* means that the geometry is from time step 1 and the color is from time step 3). The correspondences are defined by the scene flow computed from SDF flow.
>
> **Long-term scene flow evaluation.** We now provide the full optical flow evaluation results on the CMU Panoptic dataset and the long-term scene flow evaluation results on the synthetic sequence (up to future 5 frames).
>
> Since we cannot obtain the ground truth scene flow for real-world sequences of the CMU Panoptic dataset, we try to evaluate the scene flow with optical flow by projecting it onto the image plane. In the table below, we provide such evaluation results on all 5 sequences of the CMU Panoptic dataset. The optical flow computed from our scene flow is consistently better than that of NDR.
>
> | EPE (pixel) | Ian3 | Haggling\_b2 | Band1 | Pizza1 | Cello1 | avg |
> |---|:---:|:---:|:---:|:---:|:---:|:---:|
> | NDR (Cai et al., 2022) | 6.86 | 4.43 | 1.66 | 3.11 | 2.41 | 3.69 |
> | Ours| **3.49** | **2.97** | **1.18** | **1.18** | **1.34** | **2.04** |
>
> In addition, we follow the advice of Reviewer jfGg to directly evaluate the scene flow on the synthetic sequence shown in Figure 7 of the main script. The results are reported in the table below. We compare the results of three models: NDR (Cai et al., 2022), Ours, Ours with scene flow regularization which includes several regularization computed from SDF flow and scene flow (Please refer to the response to Reviewer nuCZ W-3 or Section A6 of the main script for more details). We also evaluate the long-term correspondences up to 5 frames. In particular, for every surface point in a frame, we compute the scene flow to the future first, second, and 5-th frames ("+1", "+2", and "+5"). We report the average end-point-error (EPE) in millimeter.  Our model without any explicitly scene flow regularization consistently outperforms NDR (Cai et al., 2022). With further regularization, the scene flow can be drastically improved.
>
> | EPE (mm) | +1 | +2 | +5 |
> |---|:---:|:---:|:---:|
> | NDR (Cai et al., 2022)| 16.8 | 34.8 | 90.2 |
> | Ours | 11.7 | 24.7 | 71.4 |
> | Ours w/ reg | **5.8** | **13.0** | **51.9** |
>
> - **Integral Evaluation**
>
> Yes, the temporal integration scales linearly with scene length. The second-order Runge-Kutta solver requires $23*2=46$ function evaluations for the 24-th frame.
>
> - **More Related Works**
>
> Thank you for the suggestion. We have updated the related work section to discuss those papers.
>
> - **Limitations**
>
> Thank you for the suggestion. We add the following discussion about the limitation of our work in the main script.
>
> Similar to all other multi-view reconstruction works, the reconstruction quality highly rely on the number of views. Another limitation of our SDF flow is the time complexity. To solve for the integral (Equation 7) for any query point, the number of function evaluation grows with respect to time. We aim to address those limitations in the future work. For the first limitation, since we now have an explicit relation between the scene geometry and scene motion, we may be able to achieve better geometry with limited number of views by seeking extra regularization from motion such as optical flow. For the second limitation, a potential solution could be that instead of integrating from the initial time step, we can integrate within a fixed time window.
>
> - **Input Assumptions**
>
> We acknowledge that the SDF itself does not have any restrictions on its temporal evolution. The continuity and differentiability of the SDF as a function of time for a given 3D point come from the assumption that the scene geometry evolves in a continuous way such as cars running on the road, and all human activities. Although such assumption does not hold when an object appears out of nowhere, we would like to argue that such case is scarcely possible in our daily life.
>
> - **Wrong Argument**
>
> Thank you for pointing it out. We have updated that paragraph as follow to avoid confusion.
>
> To extend NeRF to dynamic scenes, the most commonly adopted strategy is to jointly optimize an additional function that models the deformation in 3D space such function either maps all observation spaces to a canonical one such as (Park et al., 2021a) or models the temporal motion of the scene such as (Li et al., 2021). In the next section, we propose a drastically different representation, i.e., SDF flow, which can also model dynamic scenes.
>
> - **Labelling**
>
> Thank you for pointing it out. We have updated the Figure.

---

> > ### Comment · Reviewer_uMrz · 2023-11-20
> > **Thank you**
> >
> > Thank you for the response, it is very detailed and addresses almost everything sufficiently for me, including the concerns of the other reviewers. I interpret the new results as showing that the extracted scene flow quickly drifts within a few timesteps and that it is fairly inaccurate even for a single timestep. Both these downsides are common in the literature, as far as I know, and so this is not an issue from my side at least. Unless further discussion changes my mind somehow about the submission, I will increase my rating to accept if my remaining main concern is addressed.
> >
> > Main remaining concern:
> >
> > The third point under Weaknesses by reviewer jfGg is a very valuable ablation to try. The rebuttal addresses this via an experiment about temporal interpolation when using SDF Flow vs. a time-dependent SDF s(x,t). I have three concerns here: (1) It doesn't cover the reviewer's suggestion to directly predict the integral, i.e. decomposing SDF(x,t) = SDF(x,0) + delta(x,t) where both functions are learned. This would allow for constant-time evaluation rather than linear-time evaluation of the SDF. (Why decompose at all? That's what SDF Flow also does.) (2) I'm not following why temporal interpolation is used here. Why not straightforward reconstruction quality as in Table 1 of the main PDF? (3) Will this be added to the appendix?
> >
> > Minor points about the writing:
> >
> > Given that experiments in the rebuttal show that further regularization and optical-flow supervision improve the derived scene flow, I believe that the scene flow is noisy (which is to be expected) and that it benefits from appearance cues beyond pure geometry cues (an ablation would be necessary to confirm this). I'd much appreciate it if the paper could state in the introduction already that this is a limitation, maybe if only a short phrase referring to a longer sentence in the appendix.
> >
> > Then I'd like to ask to please modify the statement about topology change in the Abstract. NSFF, for example, also handles topology changes without any extra effort since it uses a naively time-conditioned dynamic NeRF.
> >
> > Please also add the sentences from [Input Assumptions] to the paper. This could be a short phrase like "real world" or "realistic" or some such, potentially with details in the appendix. It currently still sounds like it's true in mathematical generality, which it is not.
> >
> > Please add the response to reviewer nuCZ's "W-2. Validity on the assumption 2." to the appendix, it is a much clearer description than what is currently in the paper.

---

> > > ### Author Response · Authors · 2023-11-21
> > > **Thank you for the responses**
> > >
> > > We thank the reviewer for all the valuable comments that help to improve the paper.
> > >
> > > - **Results of directly predicting the integral**
> > >
> > > We provide the results of directly predicting the integral (named "Ours w/ Pred. Int.") in the table below. The performance gain of SDF flow is because the integral used to compute SDF further introduces temporal dependencies across different frames. As also envidenced by the temporal interpolation results where the frame is unseen during training, the model purly relies on the learned temporal cues to do the reconstruction on those frames.
> > >
> > > |  | Reconstruction |  |  |  | Interpolation |  |  |
> > > |---|:---:|:---:|:---:|:---:|:---:|:---:|:---:|
> > > |  | acc | comp | overall |  | acc | comp | overall |
> > > | Ours w/ SDF | 12.4 | 19.0 | 15.7 |  | 13.0 | 20.8 | 16.9 |
> > > | Ours w/ Pred. Int. | 11.8 | 19.4 | 15.6 |  | 12.3 | 22.9 | 17.6 |
> > > | Ours w/ SDF flow | **11.6** | **18.9** | **15.2** |  | **11.5** | **18.9** | **15.2** |
> > >
> > > - **Why temporal interpolation**
> > >
> > > We would like to use the temporal interpolation results to show that given multi-view videos, using SDF flow better captures the temporal depenences of the scene geometry than estimating the per-frame SDF. As shown in the above table, there is a clear gap between the resconstruction and interpolation results for the per-frame SDF models ("Ours w/ SDF" and "Ours w/ Pred. Int.") while the performance of our model with SDF flow is consistent.
> > >
> > > - **Will this be added to the appendix**
> > >
> > > We have added the results of SDF v.s. SDF flow to the appendix.
> > >
> > > - **Minor points about the writing**
> > >
> > > Thank you for the suggestions. We have made the following updates to the script accordingly.
> > >
> > > 1. We described the noisy scene flow in the introduction and refer to the appendix for detailed evaluation.
> > >
> > > 2. We modifed the statement about topology change in the abstract section.
> > >
> > > 3. We added "in the real world scenario" in the paragraph describing the continuity and differentiability of the SDF.
> > >
> > > 4. We added the detailed description about assumption 2 to the appendix.

---

> > > > ### Comment · Reviewer_uMrz · 2023-11-21
> > > > **Thank you again**
> > > >
> > > > This makes sense. Thank you for all the effort with the additional experiments and changes in writing! I have increased my score to accept (8).

---

### Official Review · Reviewer_hwJN · 2023-10-30

**Soundness:** 3 good
**Presentation:** 3 good
**Contribution:** 3 good
**Rating:** 8
**Confidence:** 4

**Summary:**

This paper proposes a novel representation, namely SDF Flow, for solving multi-view dynamic scene reconstruction by representing the dynamic scene as a 4-D SDF field and modelling its derivative w.r.t. time instead of the SDF value. The proposed representation has several nice properties and can be used to compute the scene flow analytically. Experimental results on public datasets have shown competitive reconstruction quality compared to state-of-the-art, and better scene flow estimation.

**Strengths:**

1. The proposed representation is novel. I’ve never seen similar representations before. The idea is clear and elegant but effective. It has the potential to open up a new paradigm for dynamic scene reconstruction and provide great insight for the community.

2. The authors also revealed the relationship between SDF flow and scene flow through mathematical derivation, and have shown that the scene motion can be computed analytically by solving linear equations.

3. Quantitative and qualitative experiments have shown the proposed method could achieve promising reconstruction quality. The scene flow estimation result also looks promising.

**Weaknesses:**

1. In Sec 3.3, the authors made two assumptions when deriving the computation of scene motion from SDF flow, which is justified by a 2D toy example. However, it would be great to have a more in-depth theoretical analysis and experiments to understand its convergence behaviour. Especially for assumption 2, the difference between assumed $\Delta_S$ and real value looks quite big.

2. The optimisation takes too long, which limits its practical usage.

**Questions:**

Please see weakness.

---

> ### Author Response · Authors · 2023-11-19
> **Responses to Reviewer hwJN**
>
> - **Theoretical analysis about the assumption**
>
> For assumption 1, it assumes the scene to be reconstructed is piece-wise rigid which is a common assumption in modeling scene motion  (Vogel et al., 2013).
>
> For assumption 2, we provide a more detailed description as follow.
>
> Let us denote a surface point as $\\mathbf{x}$, its corresponding point on the evolved surface as $\\mathbf{x}'$, the normal of the tangent plane at $\\mathbf{x}'$ as $\\mathbf{n}(\\mathbf{x}')$, and the closest point on the evolved surface to $\\mathbf{x}$ as $\\mathbf{x}''$. According to the definition of SDF, the absolute SDF change of $\\mathbf{x}$ i.e., $|\\Delta s|$ is equal to $||\\mathbf{x}-\\mathbf{x}''||_2$. With the assumption that the infinitely small surface region around $\\mathbf{x}$ as well as its deformation is smooth, as the time period $\\Delta t$ approaches zero, $\\mathbf{x}''$ becomes infinitely close to $\\mathbf{x}'$ which can be regarded as lying on the tangent plane to the evolved surface at $\\mathbf{x}'$. So that the absolute SDF change $|\\Delta s|$ of $\\mathbf{x}$ will equal to the distance from $\\mathbf{x}$ to such tangent plane i.e., $|(\\mathbf{x}'-\\mathbf{x})^T\\mathbf{n}(\\mathbf{x}')|$. The sign of $\\Delta s$ is determined by angle between the scene flow and the normal of the tangent plane.
>
> We have updated the Figure 3 for better visual explanation of the assumption.
>
> - **Time consuming**
>
> Thank you for pointing it out. We acknowledge that the time complexity is one of our limitations and will try to improve it in our future research. We also add the following discussion about the limitation in the script.
>
> Similar to all other multi-view reconstruction works, the reconstruction quality highly rely on the number of views. Another limitation of our SDF flow is the time complexity. To solve for the integral (Equation 7) for any query point, the number of function evaluation grows with respect to time. We aim to address those limitations in the future work. For the first limitation, since we now have an explicit relation between the scene geometry and scene motion, we may be able to achieve better geometry with extra regularization from motion such as optical flow. For the second limitation, a potential solution could be that instead of integrating from the initial time step, we can integrate within a fixed time window.
>
> **References**
>
> Christoph Vogel, Konrad Schindler, and Stefan Roth. Piecewise rigid scene flow. In Proceedings of the IEEE International Conference on Computer Vision, pp. 1377–1384, 2013.

---

> > ### Comment · Reviewer_hwJN · 2023-11-21
> >
> > I would like to thank the authors for their responses to my questions. After reading their responses and other reviewers' comments, I will keep my initial rating of Accept.

---

### Official Review · Reviewer_jfGg · 2023-10-30

**Soundness:** 3 good
**Presentation:** 2 fair
**Contribution:** 2 fair
**Rating:** 8
**Confidence:** 4

**Summary:**

This paper proposes to predict the first-order derivative of a signed distance function (SDF) at a point, termed as SDFflow. By using SDFlow, the dynamic recontruction to recover a SDF at time t can be treated as space-time integration of the SDFflow from time t_0 to t and its starting SDF at time t_0. The authors further demonstrate how locally rigid scene-flow can be recovered least-square optimization given the locally rigid assumption for small motions. The results demonstrate the method is outperforming previous baselines using alternative shape and flow representation.

**Strengths:**

* The approach is technically sound. It is kinda intuitive that this representation should work.
* From the quantitative comparison, it is clear the method is outperforming previous methods by a relatively large margin.
* The author also provides connection of SDF flow to scene flow with a math derivation. It is helpful for readers to understand its connection if without background in optimization based motion estimation.

**Weaknesses:**

* The flow evaluation in the paper is very weak. The method only compares to NDR on 2D project error using RAFT as pesudo ground truth. This should not be hard to achieve if using a synthetic dataset, if using the rigid moving toy example being shown in the paper.
* The flow estimation also lacks details. Solving the least-square optimization requires sample multiple points, which the author say "we select more than 6 points" and sometimes it still be ill-conditioned depending on the property of the sampled points, but I don't see any discussion related to this and how the results are generated.
* The main contribution of this paper is the SDFlow and a claim that it is beating alternative representation (SDF with warping field for example). This can be much clearer if the authors can provide more ablations studies on the contrast of the two representations. Two small experiments they author can simply do is to 1) have the network predict the integral of flow in eq. (7), or 2) predict the time dependent SDF s(x, t), all with fixed hyper-parameters. The contrasts in this ablation can reflect the performance difference in predicting SDF flow. Though the authors provide comparisons to alternative papers that the other papers compound too many other terms, and I don't think I can get the insights of why SDF flow works better here.

**Questions:**

1. Though the method is outperforming previous methods by big margin in the qualitative comparisons, the difference in qualitative comparison is less clear to me from all figures in the paper. In particular compared to Tensor4D, I can see each method wins in different level of details being captured. The paper currently summarizes it as "all of its results share similar artifacts which we conjecture are due to the tensor decomposition. Our method performs comparably to the baselines, with fewer artifacts." I am not sure I can capture this from the current presentation. Will love to know a concrete summary of the areas where the model increase performance best.
2. I did not see any discussions about the limitations of existing methods or representations. As all current multi-view dynamic reconstruction work, I assume this work will face the same limitations in  large motion (which will also break the linear motion relation between scene flow and sdf flow) and number of views. But I'd like to some more technical insights of potential downside using SDF flow in some scenarios compared to other representation.

---

> ### Author Response · Authors · 2023-11-19
> **Responses to Reviewer jfGg (Part 1)**
>
> - **Scene flow evaluation**
>
> Thank you for the suggestion. We now provide the full optical flow evaluation results on the CMU Panoptic dataset and the scene flow evaluation results on the synthetic sequence.
>
> Since we cannot obtain the ground truth scene flow for real-world sequences of the CMU Panoptic dataset, we try to evaluate the scene flow with optical flow by projecting it onto the image plane. In the table below, we provide such evaluation results on all 5 sequences of the CMU Panoptic dataset. The optical flow computed from our scene flow is consistently better than that of NDR.
>
> | EPE (pixel) | Ian3 | Haggling\_b2 | Band1 | Pizza1 | Cello1 | avg |
> |---|:---:|:---:|:---:|:---:|:---:|:---:|
> | NDR (Cai et al., 2022) | 6.86 | 4.43 | 1.66 | 3.11 | 2.41 | 3.69 |
> | Ours| **3.49** | **2.97** | **1.18** | **1.18** | **1.34** | **2.04** |
>
> In addition, we directly evaluate the scene flow on the synthetic sequence shown in Figure 7 of the main script. The results are reported in the table below. We compare the results of three models: NDR (Cai et al., 2022), Ours, Ours with scene flow regularization includes several regularization computed from SDF flow and scene flow (Please refer to the response to Reviewer nuCZ W-3 or Section A6 of the main script for more details). We also evaluate the long-term correspondences up to 5 frames. In particular, for every surface point in a frame, we compute the scene flow to the future first, second, and 5-th frames ("+1", "+2", and "+5"). We report the average end-point-error (EPE) in millimeter.  Our model without any explicitly scene flow regularization consistently outperforms NDR (Cai et al., 2022). With further regularization, the scene flow can be drastically improved.
>
> | EPE (mm) | +1 | +2 | +5 |
> |---|:---:|:---:|:---:|
> | NDR (Cai et al., 2022)| 16.8 | 34.8 | 90.2 |
> | Ours | 11.7 | 24.7 | 71.4 |
> | Ours w/ reg | **5.8** | **13.0** | **51.9** |
>
> - **Details about the scene flow optimization**
>
> Thank you for the comments, we clarify the details as below and add them to the Section A2 of the main script.
>
> Given a set of points $\\mathcal{P}$ sampled uniformly from a reconstructed mesh, for any point $\\mathbf{x}$ in this set, we first obtain its $K$ nearest neighbours $\\{\\mathbf{y}_1,\\mathbf{y}_2,\cdots,\\mathbf{y}_K\\}$, where $\\mathbf{y}_i\\in\\mathcal{P}$. For every neighbouring points $\\mathbf{y}_i$, we query its surface normal $\\mathbf{n}(\\mathbf{y}_i)$ and SDF flow $d(\\mathbf{y}_i)$ from the MLPs. According to Equation 17, we obtain a matrix $\\mathbf{A}\\in\\mathbb{R}^{K\\times 6}$ where the $i$-th row $\\mathbf{A}_i=[(\\mathbf{y}_i\\times\\mathbf{n}(\\mathbf{y}_i))^T,\\mathbf{n}(\\mathbf{y}_i)^T]$ and the SDF flow vector $\\mathbf{d}\\in\\mathbb{R}^K$ where $\\mathbf{d}_i=d(\\mathbf{y}_i)$. We can then solve such problem as
>
> $$
> \begin{aligned}
>     \begin{bmatrix}\\boldsymbol\\omega^*\\\\ \\mathbf{v}^*\end{bmatrix}
>     &= \text{arg min}_{\\boldsymbol\\omega, \\mathbf{v}} || \\mathbf{d} + \\mathbf{A}\begin{bmatrix}
>                 \\boldsymbol\\omega\\\\
>                 \\mathbf{v}
>             \end{bmatrix} ||_2^2 \\\\
>             &= - (\\mathbf{A}^T\\mathbf{A})^{-1}\\mathbf{A}^T\\mathbf{d}
> \end{aligned}
> $$
> where $\boldsymbol\omega^*, \mathbf{v}^*$ is the optimal angular velocity and velocity.
>
> In practice, we uniformly sample around $2000$ points from a reconstructed mesh and choose around $200$ neighbouring points to compute the scene flow of one surface point. To further study the influence of the number of neighbouring point, we compare the scene flow error with respect to the number of neighbouring points in the table below.
>
> | K | 50 | 100 | 200 | 500 | 1000 |
> |---|:---:|:---:|:---:|:---:|:---:|
> | EPE (mm) | 13.7 | 12.6 | 11.7 | 11.3 | 12.0 |
>
> - **SDF flow v.s. SDF**
>
> Thank you for the suggestion. We compare the reconstruction results of our model directly output the SDF and SDF flow. During training we set aside several frames to evaluate the interpolation ability of both representations. For example, we train the model on $\\{1,4,7,\\cdots\\}$-th frames and the evaluation results on those frames are refer to as reconstruction and those on $\\{2,3,5,6,\\cdots\\}$-th frames are refer to as interpolation. Our model with SDF flow representation consistently outperforms that with SDF representation especially at the unseen frames (interpolation).
>
> |  | Reconstruction |  |  |  | Interpolation |  |  |
> |---|:---:|:---:|:---:|:---:|:---:|:---:|:---:|
> |  | acc | comp | overall |  | acc | comp | overall |
> | Ours w/ SDF | 12.4 | 19.0 | 15.7 |  | 13.0 | 20.8 | 16.9 |
> | Ours w/ SDF flow | **11.6** | **18.9** | **15.2** |  | **11.5** | **18.9** | **15.2** |

---

> > ### Author Response · Authors · 2023-11-19
> > **Responses to Reviewer jfGg (Part 2)**
> >
> > - **Questions**
> >
> > **1. Detailed qualitative comparison to Baselines**
> >
> > The artifacts refer to as the check board effects on the surface of the mesh such as the thigh of the first sample and the dress of the second sample. We have updated the main script to highlight those regions.
> >
> >
> > **2. Limitations**
> >
> >  Thank you for the suggestion. We add the following discussion about the limitation of our work in the main script and clarify about the large motion below.
> >
> > Similar to all other multi-view reconstruction works, the reconstruction quality highly rely on the number of views. Another limitation of our SDF flow is the time complexity. To solve for the integral (Equation 7) for any query point, the number of function evaluation grows with respect to time. We aim to address those limitations in the future work. For the first limitation, since we now have an explicit relation between the scene geometry and scene motion, we may be able to achieve better geometry with limited number of views by seeking extra regularization from motion such as optical flow. For the second limitation, a potential solution could be that instead of integrating from the initial time step, we can integrate within a fixed time window.
> >
> > We would also like to clarify that as long as the surface and the motion are smooth, even large motion will not break the relation between the scene flow and SDF flow. Because the derivation of such relation is the limit as time approaches zero.

---

> > > ### Comment · Reviewer_jfGg · 2023-11-20
> > >
> > > Thanks for providing the extra experiments and explanations to address my questions.
> > >
> > > Overall the paper has been revised to a good state which I think quality a good submission. I suggest one the of flow evaluations can be moved to main paper.

---

> > > > ### Author Response · Authors · 2023-11-21
> > > > **Thank you for the responses**
> > > >
> > > > We thank the reviewer for all the valuable comments that help to improve the paper.
> > > >
> > > > We have moved the full optical flow evaluation results on the CMU Panoptic dataset to the main paper, and added the results of SDF v.s. SDF flow to the Appendix.

---

### Official Review · Reviewer_nuCZ · 2023-10-31

**Soundness:** 3 good
**Presentation:** 2 fair
**Contribution:** 2 fair
**Rating:** 8
**Confidence:** 4

**Summary:**

This paper proposes an algorithm for implicit surface reconstruction in dynamic scenes. The key is to bridge the SDF flow and sceneflow by assuming the two hypothesis. Also, typically this paper requires RGBD data which is to sample points nearby surface and to make this algorithm feasible. Accordingly, this paper could be understood as 4D surface tracker.

**Strengths:**

This paper brilliantly bridge the concept of sceneflow and the proposed SDF flow. Based on the two assumptions, one for rigidity and the other for the linearlization (?), the proposed SDF flow is differentiably convertible into the sceneflow, which enables the network to encode sceneflow from RGBD frames. As far as my understanding, this is the pioneering work. For experiment parts, comparison with NDR (Neurips 2022) is reasonable.

**Weaknesses:**

W-1. Low fidelity results

Despite the novel idea, the reconstruction quality is far below my expectation. Even though the results from NDR (Neurips 2022) are also not that good enough, there are not that much dramatic change after the authors applied the proposed SDF flow.

Of course, I can see the sceneflow visualization in Fig8 of the manuscript, the reconstruction quality is not really good enough. Moreover, __we cannot judge whether the quality of the sceneflow is correct or not.__

W-2. Validity on the assumption 2.

Can the authors further elaborate the validity of the proposed 2nd assumption? What geometric insight reside within this hypothesis?

_"[Assumption 2] As the time period Δt approaches zero, the absolute SDF change |Δs| of a surface point x equals the distance from x to the tangent plane to the evolved surface at the corresponding point x′ and the sign of Δs is determined by the angle between that tangent plane’s normal and the scene flow (as shown in Figure 3)."_

W-3. Necessity of Sec 3.3

While training the network, does the understanding of sec 3.3 is needed? While this paper proposes an algorithm for bridging the SDF flow and sceneflow, there are not much material or experiments that clearly demonstrate the accuracy of the sceneflow. Moreover, the proposed assumptions are not used when training the methods. Accordingly, it is quite confusing me to understand the precise pipeline of this paper.

**Questions:**

Please refer to the question above. Especially for W-3, if there are some things that I misunderstood, please let me know.

Overall, I am quite positive to this paper. Depending on the rebuttal, let me change my score.

**Details Of Ethics Concerns:**

This paper does not require ethics review.

---

> ### Author Response · Authors · 2023-11-19
> **Responses to Reviewer nuCZ**
>
> - **W-1 Low fidelity results**
>
> The reason for low fidelity results is because of the challenging data. We try to reconstruct the geometry of a dynamic scene with only 10 sparse views. In Section A5, Figure 17, we show all the 10 views at particular time step. The view changes are very large making it extreme hard to reconstruct the geometry not only for NeRF-based methods but also for other multi-view stereo ones. In fact, even COLMAP (Sch ̈onberger & Frahm, 2016) fails on those images due to “no good initial image pair found”.
>
> **Scene flow evaluation.** Since we cannot obtain the ground truth scene flow for real-world sequences of the CMU Panoptic dataset, we try to evaluate the scene flow with optical flow by projecting it onto the image plane. In the table below, we provide such evaluation results on all 5 sequences of the CMU Panoptic dataset. The optical flow computed from our scene flow is consistently better than that of NDR.
>
> | EPE (pixel) | Ian3 | Haggling\_b2 | Band1 | Pizza1 | Cello1 | avg |
> |---|:---:|:---:|:---:|:---:|:---:|:---:|
> | NDR (Cai et al., 2022) | 6.86 | 4.43 | 1.66 | 3.11 | 2.41 | 3.69 |
> | Ours| **3.49** | **2.97** | **1.18** | **1.18** | **1.34** | **2.04** |
>
> In addition, we follow the advice of Reviewer jfGg to directly evaluate the scene flow on the synthetic sequence shown in Figure 7 of the main script. The results are reported in the table below. We compare the results of three models: NDR (Cai et al., 2022), Ours, Ours with scene flow regularization which we will introduce in the response to W-3. We also evaluate the long-term correspondences up to 5 frames. In particular, for every surface point in a frame, we compute the scene flow to the future first, second, and 5-th frames ("+1", "+2", and "+5"). We report the average end-point-error (EPE) in millimeter.  Our model without any explicitly scene flow regularization consistently outperforms NDR (Cai et al., 2022). With further regularization, the scene flow can be drastically improved.
>
> | EPE (mm) | +1 | +2 | +5 |
> |---|:---:|:---:|:---:|
> | NDR (Cai et al., 2022)| 16.8 | 34.8 | 90.2 |
> | Ours | 11.7 | 24.7 | 71.4 |
> | Ours w/ reg | **5.8** | **13.0** | **51.9** |
>
> - **W-2. Validity on the assumption 2.**
>
> Let us denote a surface point as $\\mathbf{x}$, its corresponding point on the evolved surface as $\\mathbf{x}'$, the normal of the tangent plane at $\\mathbf{x}'$ as $\\mathbf{n}(\\mathbf{x}')$, and the closest point on the evolved surface to $\\mathbf{x}$ as $\\mathbf{x}''$. According to the definition of SDF, the absolute SDF change of $\\mathbf{x}$ i.e., $|\\Delta s|$ is equal to $||\\mathbf{x}-\\mathbf{x}''||_2$. With the assumption that the infinitely small surface region around $\\mathbf{x}$ as well as its deformation is smooth, as the time period $\\Delta t$ approaches zero, $\\mathbf{x}''$ becomes infinitely close to $\\mathbf{x}'$ which can be regarded as lying on the tangent plane to the evolved surface at $\\mathbf{x}'$. So that the absolute SDF change $|\\Delta s|$ of $\\mathbf{x}$ will equal to the distance from $\\mathbf{x}$ to such tangent plane i.e., $|(\\mathbf{x}'-\\mathbf{x})^T\\mathbf{n}(\\mathbf{x}')|$. The sign of $\\Delta s$ is determined by angle between the scene flow and the normal of the tangent plane.
>
> - **W-3. Necessity of Sec. 3.3.**
>
> We acknowledge that currently the training of our model does not involve the computation of scene flow. However, bridging the gap between SDF flow and scene flow could potentially be applied to train a better model. We evidence this by the toy example shown in Figure 7 of the main script. In particular, we train a new model (named ``Ours w/ reg'') that outputs not only the SDF flow but also the scene flow, we then apply several priors:
>
> 1. SDF flow and scene flow consistency prior from Equation 16.
>
> 2. Piece-wise rigid prior: we encourage the scene flow of close points to be similar.
>
> 3. Optical flow loss: we project the scene flow onto image plane to obtain the optical flow and compute optical flow loss with it.
>
> More details about this experiment are added to Section A6 of the main script. The evaluation results are show in the table below. The scene flow regularization helps to improve the reconstruction quality. As also evidenced in the response to W-1, the estimated scene flow from SDF flow is also improved. We believe revealing the mathematical relation between the scene motion and scene geometry has more potential applications which we will explore in the future.
>
> | (mm) | acc | comp | overall |
> |---|:---:|:---:|:---:|
> | NDR (Cai et al., 2022) | 33.5 | 25.3 | 29.4 |
> | Ours | 31.8 | 23.3 | 27.6 |
> | Ours w/ reg | **27.2** | **19.1** | **23.2** |
>
> **References**
>
> Johannes Lutz Sch ̈onberger and Jan-Michael Frahm. Structure-from-Motion Revisited. In Conference on Computer Vision and Pattern Recognition (CVPR), 2016.

---

> > ### Comment · Reviewer_nuCZ · 2023-11-21
> > **Thank you for the rebuttal.**
> >
> > Overall, most of my concerns are properly discussed. Typically, the answer for W-2 is really impressive. I really enjoyed reading the geometric insight of the authors on dynamic surface understanding.
> >
> > Before I increase my final rate, I have one more question regarding the comments by the reviewer jfGg in weakness section, typically for the third weakness
> >
> > - _The main contribution of this paper is the SDFlow and a claim that it is beating alternative representation (SDF with warping field for example). This can be much clearer if the authors can provide more ablations studies on the contrast of the two representations._ ... __and I don't think I can get the insights of why SDF flow works better here__.
> >
> > As the two reviewers (jfGg and uMrz) commented in their review, can the authors further elaborate why the SDF flow estimation brings much higher fidelity results than the quality of direct SDF estimation? I checked the quantitative / qualitative results. However, the clear insight is the one that I want to clarify.
> >
> > __change rate: 5 -> 8: accept (after the authors reply my last question)__

---

> ### Author Response · Authors · 2023-11-21
> **Thank you for the responses**
>
> We thank the reviewer for all the valuable comments that help to improve the paper.
>
> - **The reason why SDF flow is better**
>
> We think it is because that SDF flow brings temporal regularization to the SDF. Given multi-view videos, one can directly predict the per-frame SDF. However, since each frame are predicted almost independently (using the same function with time as an extra input), the temporal information of the scene is not fully utilized. By contrast, SDF flow models the change of the scene geometry which captures the temporal dependencies across different frames via the integral defined in Equation 7. For example, it is hard for the pre-frame SDF to recover an occluded object while it is possible for SDF flow to reconstuct it if it is observed in future or/and past frames. Similarly, for fast moving surfaces that often lead to motion blur on the images, it is hard for SDF to recover the correct geometry given only such blurry images. However, the temporal information introduced by SDF flow could be very helpful in this case.

---

> > ### Author Response · Authors · 2023-11-23
> >
> > Dear Reviewer nuCZ,
> >
> > Thank you again for the comments.
> >
> > It is towards the end of the discussion phase. If you have any futher concerns, please let us know. We will try to address them as soon as possible.
> >
> > Kind regards,
> >
> > Paper 6442 authors

---

### Meta-Review · Area_Chair_Hubr · 2023-12-07

**Metareview:**

The submission received positive reviews from all the reviewers (after extensive discussions with the authors). The reviewers appreciates the elegance of the method, connection between scene flow and SDF representations, and strong results demonstrated. Initially, reviewers have some concerns about the evaluation and quality of results, but they have been subsequently addressed throughout the discussions. After reading the paper, the reviewers' comments, the authors' rebuttal and the discussions, the AC agrees with the decision by the reviewers and recommends acceptance.

**Justification For Why Not Higher Score:**

I believe the significance of the results, in terms of improvements over prior works and practicality for downstream applications, is not quite sufficient to be considered for oral/spotlight presentation.

**Justification For Why Not Lower Score:**

The paper tackles a challenging problem of dynamic 3D reconstruction with an elegant formulation, and received unanimous acceptance recommendation. I believe its merits deserves acceptances.

---

### Decision · Program_Chairs · 2024-01-16

Accept (poster)